# HIGH-POWER TRAINING DATA IDENTIFICATION WITH PROVABLE STATISTICAL GUARANTEES

## ABSTRACT

Identifying training data within large-scale models is critical for copyright litigation, privacy auditing, and ensuring fair evaluation. The conventional approaches treat it as a simple binary classification task without statistical guarantees. A recent approach is designed to control the false discovery rate (FDR), but its guarantees rely on strong, easily violated assumptions. In this paper, we introduce Provable Training Data Identification (**PTDI**), a rigorous method that identifies a set of training data with strict false discovery rate (FDR) control. Specifically, our method computes p-values for each data point using a set of known unseen data, and then constructs a conservative estimator for the data usage proportion of the test set, which allows us to scale these p-values. Our approach then selects the final set of training data by identifying all points whose scaled p-values fall below a data-dependent threshold. This entire procedure enables the discovery of training data with provable, strict FDR control and significantly boosted power. Extensive experiments across a wide range of models (LLMs and VLMs), and datasets demonstrate that PTDI strictly controls the FDR and achieves higher power. Our implementation code is available at https://anonymous.4open.science/r/Provable_Training_Data_Identification-408E.

## 1 INTRODUCTION

The extensive deployment of machine learning models has driven an ongoing demand for large-scale datasets, which has raised significant legal challenges, including copyright disputes (Bartz et al., 2024; Disney Enterprises, Inc., 2025), data privacy concerns (European Parliament & Council of the European Union, 2016; California State Assembly, 2018), and issues of data contamination from evaluation benchmarks (Sainz et al., 2023; Balloccu et al., 2024). These concerns raise the importance of identifying a specific, well-defined set of data allegedly used in training. For instance, Strike 3 alleges that Meta infringed on at least 2,396 of its copyrighted films in its lawsuit (Strike 3 Holdings, LLC and Counterlife Media, LLC, 2025), a claim with potential statutory damages exceeding $350 million. To resolve such high-stakes disputes, claims must be supported by credible evidence that strictly controls the risk of false positives. This underscores the need for methods that provide rigorous statistical guarantees for identifying training data.

To this end, prior studies (Shi et al., 2024; Li* et al., 2024; Zhang et al., 2025a) have developed various methods to detect training data in large language models (LLMs) and vision-language models (VLMs). These methods typically rely on computing a prediction score (e.g., perplexity or entropy) for level-set estimation without theoretical guarantees. A recent approach (Hu et al., 2025) proposes to construct knockoff statistics to control the false discovery rate (FDR) (i.e., the expected proportion of false positives among the identifications). However, this approach requires access to model gradients, which are unavailable in black-box settings. Additionally, the difficulty of constructing effective knockoffs makes it susceptible to violating its assumption of a symmetrically distributed statistic, resulting in invalid FDR control. These limitations motivate us to design a distribution-free method that provides rigorous FDR control under both white-box and black-box settings.

In this work, we introduce Provable Training Data Identification (**PTDI**), a novel method to discover training data with a strict FDR guarantee. Our approach begins by computing a detection score (e.g., perplexity) for each data point and then uses a non-training calibration set to construct p-

values through conformal inference. To improve power, we mitigate the conservativeness of the Benjamini–Hochberg (BH) procedure by introducing a subtraction estimator that approximates the data usage proportion to scale the p-values. The scaled p-values are then passed to the BH procedure (Benjamini & Hochberg, 1995; Benjamini & Yekutieli, 2001) to yield the final identified training set. We provide rigorous theoretical guarantees that PTDI strictly controls the FDR with distribution-free guarantees. The resulting approach is model-agnostic and applicable in both white-box and black-box settings, while easily integrating with existing detection methods, making it a practical tool for training data identification.

We empirically validate our method through extensive experiments across diverse settings, including pre-training and fine-tuning paradigms for large language models (LLMs) and vision-language models (VLMs) on various datasets. Across all settings, our method consistently controls the realized FDR below the target level, providing strong empirical support for our theoretical guarantees. For instance, on WikiMIA (Shi et al., 2024) with Pythia-1.4B (Biderman et al., 2023) at a target FDR of 0.05, our method achieves an empirical FDR of 0.0494, whereas the approach of Hu et al. (2025) yields 0.1311, resulting in invalid FDR control. Moreover, we also validate that our scaling p-value procedure indeed improves power. Specifically, on WikiMIA with GPT-NeoX-20B (Black et al., 2022) at a target FDR of 0.5, our method improves power from 0.4441 to 0.7470 using the MIN-K% detection score (Shi et al., 2024).

We summarize our contributions as follows:

1. We propose **Provable Training Data Identification (PTDI)**, a novel and versatile method that achieves distribution-free, strict FDR control for training data identification, requiring only unseen data as a calibration set. Our method can be readily combined with detection scores derived from either black-box outputs or white-box gradients.

2. We provide a rigorous theoretical proof establishing that PTDI strictly controls the false discovery rate (FDR). Our analysis formally shows that the proposed data-dependent p-value scaling maintains the target FDR guarantee.

3. We conduct extensive experiments across a diverse range of models (LLMs and VLMs), tasks (pre-training and fine-tuning) and datasets. The results empirically validate that PTDI strictly controls the FDR and achieves higher power.

## 2 PRELIMINARY

### 2.1 BACKGROUND

**Training data detection.** Given a data point $X$ and a target model $\theta$ trained on dataset $\mathcal{D}_{\text{train}}$, training data detection aims to detect whether $X$ is a part of the training set $\mathcal{D}_{\text{train}}$. This problem is an instance of the membership inference attacks (MIAs) (Shokri et al., 2017), but generally applied to the LLM/VLM scenarios (Carlini et al., 2021; Zhang et al., 2024; Shi et al., 2024; Li et al., 2024b). This task is typically formulated as a binary classification problem, where the predicted label $\widehat{M} \in \{0, 1\}$ indicates whether $X$ is predicted as a training sample ($\widehat{M} = 1$) or not ($\widehat{M} = 0$). Formally, this prediction is made through level-set estimation:

$$\widehat{M} = \mathbb{1}\{T(X; \theta) \leqslant \tau\}, \tag{1}$$

where $T(X; \theta)$ denotes the detection score (e.g., perplexity) calculated from the model $\theta$, and $\tau$ is a threshold determined by a validation set. By convention, a lower detection score $T$ suggests $X$ is more likely to be trained by the target model and vice versa.

To provide a concrete understanding of the detection score $T(X; \theta)$ from Equation (1), we now introduce several widely-used examples. For LLMs, a data point $X$ is a text sequence, which we denote as $X = \{x_1, \ldots, x_L\}$. A common detection score is perplexity (Li, 2023):

$$\text{Perplexity}(X; \theta) = \exp[-\sum_{i=1}^{L} \log p_\theta(x_i \mid x_{<i})], \tag{2}$$

where $x_{<i} = (x_1, \ldots, x_{i-1})$ and $p_\theta(x_i \mid \ldots)$ is the conditional probability of token $x_i$ given the preceding tokens. A lower perplexity suggests the sequence is more familiar to the model, indicating the model was more likely trained on this data.

As for VLMs, where data points are formulated as a token sequence that contains the concatenation of token sequences from image and text, a powerful choice is based on Rényi entropy (Rényi, 1961):

$$\text{Rényi}(X;\theta) = -\frac{1}{L}\sum_{i=1}^{L}\log\{\sum_{v\in\mathcal{V}}[p_\theta(v\mid x_{<i})]^\gamma\} \tag{3}$$

where $\mathcal{V}$ is the model's vocabulary and $\gamma$ is a hyperparameter. A lower entropy score indicates the examples are more likely to have been trained by the target model. In our experiments, we follow the approach of Li et al. (2024b) and use **MaxRényi-K%**, which is the Rényi entropy averaged over the top K% largest values for a given input $X$.

## 2.2 PROBLEM STATEMENT

The conventional approach of treating training data detection as a binary classification task for individual points often fails to provide rigorous guarantees needed in real-world scenarios (Zhang et al., 2025b). In many practical applications, the objective is not merely to classify single data points but to identify a **subset of members** from a larger collection. For instance, in data contamination research, identifying and removing sets of contaminated data points is crucial for fair model evaluation (Dong et al., 2024; Zhu et al., 2024b; Zhao et al., 2024; Gao et al., 2024). Similarly, in copyright litigation, claimants must provide a specific list of infringed works, where the ability to produce a credible set of evidence can have significant financial implications These examples show the critical need for methods that can reliably identify a set of training data points.

In this work, we focus on the problem of **training data identification**, where the objective is to construct a selection set from the test data that contains a higher proportion of true member samples. Suppose we have access to a target model $\theta$, a calibration set $\mathcal{D}_{\text{cal}}$ of size $n$ and a test set $\mathcal{D}_{\text{test}} = \{X_{n+j}\}_{j=1}^{m}$ consisting of candidate training samples. Here, $M_i \in \{0,1\}$ denotes the true membership label, where $M_i = 1$ indicates that $X_i$ was used to train $\theta$. Our goal is to select a subset of indices $\mathcal{S} \subseteq \{1,\ldots,m\}$ from the $\mathcal{D}_{\text{test}}$ such that the false discovery rate (FDR) is controlled at a user-specified level $\alpha \in (0,1)$:

$$\text{FDR} = \mathbb{E}\left[\frac{\sum_{j=1}^{m}\mathbb{1}\{M_{n+j}=0, j\in\mathcal{S}\}}{\max(|\mathcal{S}|,1)}\right] \leqslant \alpha. \tag{4}$$

At this guarantee, we also desire the selected set $\mathcal{S}$ containing true training data points as much as possible, which is quantified by power:

$$\text{Power} = \mathbb{E}\left[\frac{\sum_{j=1}^{m}\mathbb{1}\{j\in\mathcal{S}, M_{n+j}=1\}}{\max(1,\sum_{j=1}^{m}\mathbb{1}\{M_{n+j}=1\})}\right]. \tag{5}$$

It is worth noting that our formulation of training data identification is different from traditional membership inference. The latter focuses on classifying single data points, and providing theoretical guarantees for this task against large models is often intractable (Zhang et al., 2025b). In contrast, our work shifts the focus to identifying a set of training data, thereby enabling rigorous statistical error control in practice. A detailed discussion of this distinction is provided in Appendix G.

In this paper, we mainly discuss the scenario that the auditor is only able to source data that are confirmed **non-members** of the training set. The calibration set $\mathcal{D}_{\text{cal}} = \{(X_i, M_i)\}_{i=1}^{n}$ is constructed such that $M_i = 0$ for all $i$. This is a widely used assumption (Ye et al., 2022; Shi et al., 2024; Zhang et al., 2025a) since it can be satisfied by using data generated after the model's training cutoff date (e.g., recent news articles or photos) or by leveraging private, proprietary data not publicly accessible for web scraping (e.g., internal corporate documents or unreleased creative works). We proceed by constructing a hypothesis testing framework with statistical guarantees using this calibration set.

## 3 METHODOLOGY

### 3.1 PROVABLE TRAINING DATA IDENTIFICATION

To achieve the training–data identification objective in Section 2.2, we formulate the problem within the framework of multiple-hypothesis testing:

$$H_j : M_{n+j} = 0, \quad j = 1,\ldots,m, \tag{6}$$

where $M_{n+j} = 0$ denotes the null hypothesis that $X_{n+j}$ was not included in training $\theta$. Rejecting $H_j$ corresponds to identifying $X_{n+j}$ as a training member and adding its index $j$ to the selected set $\mathcal{S}$. For notational convenience, let $T_i = T(X_i; \theta)$ for $i = 1, \ldots, n + m$. We then construct the conformal p-values (Vovk et al., 2003; 2005) for each test point as:

$$p_j = \frac{1 + \sum_{i=1}^{n} \mathbb{1}\{T_i \leqslant T_{n+j}\}}{n+1}, \text{ for } j = 1, \ldots, m. \tag{7}$$

Conceptually, a smaller score $T_{n+j}$ indicates that $X_{n+j}$ is more likely a training member, resulting in a smaller p-value. To collectively test hypotheses for all test instances with controlled FDR, we employ the Benjamini-Hochberg (BH) procedure (Benjamini & Hochberg, 1995). However, the standard BH procedure is conservative as its theoretical FDR bound scales with the proportion of true null hypotheses (Storey, 2002). To improve the power, we introduce scaled p-values, which adjust for the estimated proportion of training data in the target set. The scaled p-value is defined as:

$$\tilde{p}_j = (1 - \hat{\pi}_{\text{test}})p_j, \text{ for } j = 1, \ldots, m. \tag{8}$$

where $\hat{\pi}_{\text{test}}$ is an estimate of $\pi_{\text{test}}$, the proportion of training data in the test set (i.e., $\pi_{\text{test}} = \Pr(M_{n+j} = 1)$). This estimate is obtained via a data usage proportion estimator $\mathcal{E}$ such that $\hat{\pi}_{\text{test}} = \mathcal{E}(\mathcal{D}_{\text{cal}}, \mathcal{D}_{\text{test}})$. We defer the implementation details of this estimator to Section 3.2.

With these scaled p-values, we then run the BH procedure to obtain the set of identified training data. Specifically, let $\tilde{p}_{(1)} \leqslant \tilde{p}_{(2)} \leqslant \cdots \leqslant \tilde{p}_{(m)}$ denote the sorted scaled p-values, the final set is:

$$\mathcal{S} = \{j \mid \tilde{p}_j \leqslant \frac{k^*}{m}\alpha\}, \text{ where } k^* = \max\{k \mid \tilde{p}_{(k)} \leqslant \frac{k}{m}\alpha\}. \tag{9}$$

This procedure determines a data-dependent significance threshold by identifying the largest p-value $\tilde{p}_{(k^*)}$, and then identifies all data points with scaled p-values below this adaptive threshold as significant. The full procedure is detailed in Algorithm 1.

---

**Algorithm 1** Provable Training Data Identification(PTDI)

---

**Require:** Target model $\theta$, calibration data $\mathcal{D}_{\text{cal}}$, test data $\mathcal{D}_{\text{test}}$, FDR target $\alpha \in (0, 1)$, detection score function $T(\cdot)$, data usage proportion estimator $\mathcal{E}$.
1: Compute detection scores $T_i \leftarrow T(X_i; \theta)$ for all $X_i \in \mathcal{D}_{\text{cal}} \cup \mathcal{D}_{\text{test}}$.
2: Construct p-values $p_j$ as Equation (7) for $j = 1, \ldots, m$.
3: Obtain the data usage proportion estimate $\hat{\pi}_{\text{test}} \leftarrow \mathcal{E}(\mathcal{D}_{\text{cal}}, \mathcal{D}_{\text{test}})$
4: Compute scaled p-values: $\tilde{p}_j \leftarrow (1 - \hat{\pi}_{\text{test}})p_j$ for $j = 1, \ldots, m$.
5: Sort the scaled p-values: $\tilde{p}_{(1)} \leq \tilde{p}_{(2)} \leq \cdots \leq \tilde{p}_{(m)}$.
6: Find $k^* \leftarrow \max\{k \mid \tilde{p}_{(k)} \leq \frac{k}{m}\alpha\}$.
7: **if** $k^*$ exists **then**
8:     **return** Selection set $\mathcal{S} = \{j \mid \tilde{p}_j \leq \frac{k^*}{m}\alpha\}$
9: **else**
10:     **return** $\mathcal{S} = \emptyset$.
11: **end if**

---

### 3.2 ESTIMATE DATA USAGE PROPORTION

In this part, we detail a specific implementation for the data usage proportion estimator $\mathcal{E}$ required by our main procedure in Algorithm 1. The resulting estimate $\hat{\pi}_{\text{sub}}$ will be used as $\hat{\pi}_{\text{test}}$ to scale the p-values in Equation (8). Note that the distribution of detection scores, which we denote as a random variable $T$, is a mixture of the score distributions for members and non-members. This can be expressed in terms of the probability density function $p(t)$ as:

$$p_{\text{test}}(t) = \pi_{\text{test}}p(t \mid M = 1) + (1 - \pi_{\text{test}})p(t \mid M = 0). \tag{10}$$

Our strategy is to identify a region of scores $\mathcal{R}$ that is sparsely populated by member data. Specifically, we choose $\mathcal{R}$ such that the probability of a member's score falling within it is negligible, i.e., $\int_{\mathcal{R}} p(t \mid M = 1)dt \approx 0$. The integral of the detection score from the test set over this region can be

lower-bounded as follows:

$$\int_{\mathcal{R}} p(t_{\text{test}})dt = \pi_{\text{test}} \int_{\mathcal{R}} p(t \mid M = 1)dt + (1 - \pi_{\text{test}}) \int_{\mathcal{R}} p(t \mid M = 0)dt$$

$$\geqslant (1 - \pi_{\text{test}}) \int_{\mathcal{R}} p(t \mid M = 0)dt.$$

By rearranging, we can derive a lower bound for the member proportion $\pi_{\text{test}}$:

$$\pi_{\text{test}} \geqslant 1 - \frac{\int_{\mathcal{R}} p(t_{\text{test}})dt}{\int_{\mathcal{R}} p(t \mid M = 0)dt} \tag{11}$$

A naive plug-in estimator can be formed by replacing the true probabilities with their empirical estimates. Formally, we define the estimator as:

$$\hat{\pi}_{\text{sub}} = 1 - \frac{\frac{1}{m+1}\left(1 + \sum_{j=1}^{m} \mathbb{1}\{T(X_{n+j}) \in \mathcal{R}\}\right)}{\frac{1}{n}\sum_{i=1}^{n} \mathbb{1}\{T(X_i) \in \mathcal{R}\}}. \tag{12}$$

We term it as a **subtraction estimator** because it infers the member proportion by measuring how their presence effectively "subtracts" from the data density in a specific region $\mathcal{R}$ compared to a pure non-member baseline. In practice, the region $\mathcal{R} = (\tau, +\infty)$ is constructed by selecting a quantile threshold $\eta \in (0, 1)$ and identifying the score $\tau$ that partitions the calibration data accordingly.

By Equation (11), the $\hat{\pi}_{\text{sub}}$ is a conservative estimator, so it exhibits favorable property that maintains FDR control, as formalized in the following proposition.

**Proposition 1.** *Let $\hat{\pi}_{sub}$ be the subtraction estimator defined above. Assuming the test data points are i.i.d. draws from the test distribution, the expectation of the ratio of the true non-member proportion to the estimated non-member proportion is bounded by 1. Formally,*

$$\mathbb{E}\left[\frac{1 - \pi_{test}}{1 - \hat{\pi}_{sub}}\right] \leqslant 1.$$

With this proposition, we establish the following theorem:

**Theorem 1.** *Suppose the covariate of calibration set $\{X_i\}_{i=1}^{n}$ and the test set $\{X_{n+j}\}_{j=1}^{m}$ are i.i.d. Then for any $\alpha \in (0, 1)$, the selected set $\mathcal{S}$ obtained by Algorithm 1 satisfy FDR $\leqslant \alpha$. That is:*

$$\text{FDR} = \mathbb{E}\left[\frac{\sum_{j=1}^{m} \mathbb{1}\{M_{n+j} = 0, j \in \mathcal{S}\}}{\max(|\mathcal{S}|, 1)}\right] \leqslant \alpha.$$

The corresponding proofs are presented in Appendix C.1 and Appendix C.2. For reference, we also provide the evaluation of the subtraction estimator in Appendix F.1.

## 4 EXPERIMENTAL RESULTS

### 4.1 SETUP

**Models** Our experiments cover a wide range of open-source models. For Large Language Models (LLMs), we evaluate GPT-2 (Radford et al., 2019), GPT-Neo (Gao et al., 2020), GPT-NeoX-20B (Black et al., 2022), LLaMA-7B (Touvron et al., 2023), and Pythia (1.4B and 6.9B variants) (Biderman et al., 2023). For Vision-Language Models (VLMs), we use LLaVA-1.5 (Liu et al., 2023) and MiniGPT-4 (Zhu et al., 2024a).

**Datasets.** We employ six common benchmark datasets for evaluation. For LLM pre-training, we use the WikiMIA (Shi et al., 2024) and ArxivTection (Duarte et al., 2024) datasets. For fine-tuning LLMs, we utilize XSum (Narayan et al., 2018) and BBC Real Time (Li et al., 2024a). In the vision-language domain, following previous work (Li et al., 2024b), we use the VL-MIA/Flickr and VL-MIA/DALL-E datasets. The details for our experiment are presented in Appendix E.

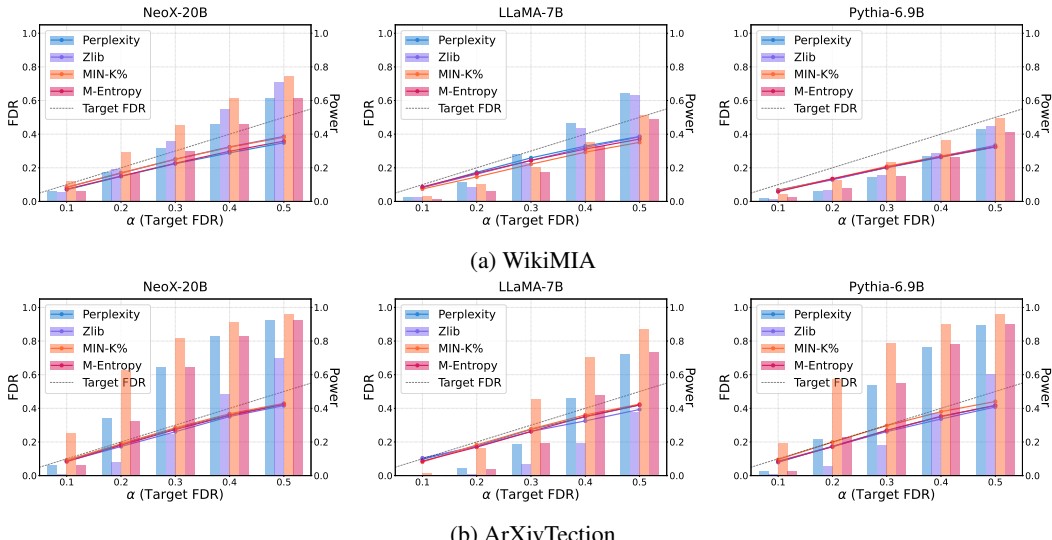

Figure 1: FDR (solid lines) and Power (bars) achieved by our method when applied to various detection scores across a range of levels $\alpha$ (target FDR ). Each subplot corresponds to the results for a specific model and dataset. The dashed diagonal line represents the desired control level $\alpha$.

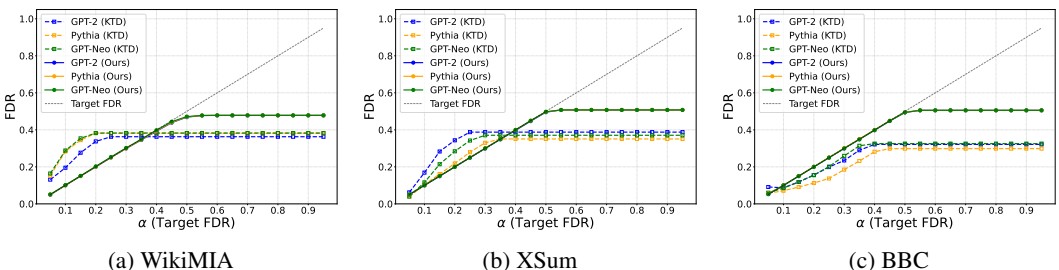

Figure 2: Comparison of FDR control between our method and KTD on three datasets.

## 4.2 MAIN RESULTS

**Our method is compatible with current pre-training data detection methods and strictly controls FDRs.** Our method is designed to be compatible with various training data detection methods. For LLMs, we demonstrate this by integrating our approach with several established scores. In addition to **Perplexity** (Li, 2023) (see Equation (2)), our experiments include the ratio of perplexity to zlib compression entropy (**Zlib**) (Carlini et al., 2021), the modified entropy (**M-Entropy**) (Song & Mittal, 2021), and **MIN-K%**, which scores a text based on the k% of its tokens with the lowest probabilities (Shi et al., 2024). The results in Figure 1 present that our method strictly and consistently controls FDR on all settings. Additional results for VLMs are provided on Appendix F.2.

**Comparison with knockoff inference-based training data detector (KTD).** We adopt the experimental setting of KTD (Hu et al., 2025), evaluating on GPT-2 (Radford et al., 2019), GPT-Neo (Gao et al., 2020), and Pythia-1.4B (Biderman et al., 2023) across the WikiMIA, XSum, and BBC Real Time datasets. To ensure a fair comparison under a white-box assumption, we set our method's detection score $T(X)$ to be the knockoff statistic from KTD. This statistic is calculated as the difference between the $L_2$ norm of the model's gradient for an input $X$ and the average $L_2$ norm of the gradients for its synthetic knockoff samples. As demonstrated in Figure 2, our method consistently maintains the target FDR across all settings, whereas KTD fails to control the FDR on WikiMIA and XSum for certain values of $\alpha$. For a complete analysis, we compare the power on BBC Real Time under conditions where KTD successfully controls the FDR ($\alpha \geqslant 0.1$), with results shown in Figure 3. The comparison reveals that our method achieves superior power on GPT-2 while performing

Table 1: Comparison of power on the WikiMIA dataset. We evaluate our method using scaled p-values (***Ours***) against the baseline using original p-values (Vanilla) across various LLMs and detection scores at different target FDR levels ($\alpha$). Higher power is highlighted in **bold**.

| Model | Method | $\alpha = 0.1$ | | $\alpha = 0.2$ | | $\alpha = 0.3$ | | $\alpha = 0.4$ | | $\alpha = 0.5$ | |
|---|---|---|---|---|---|---|---|---|---|---|---|
| | | Vanilla | ***Ours*** | Vanilla | ***Ours*** | Vanilla | ***Ours*** | Vanilla | ***Ours*** | Vanilla | ***Ours*** |
| **NeoX-20B** | Perplexity | 0.03 | **0.06** | 0.08 | **0.17** | 0.15 | **0.31** | 0.21 | **0.46** | 0.32 | **0.61** |
| | Zlib | 0.02 | **0.06** | 0.07 | **0.19** | 0.14 | **0.36** | 0.21 | **0.55** | 0.30 | **0.71** |
| | MIN-K% | 0.05 | **0.12** | 0.16 | **0.29** | 0.26 | **0.45** | 0.35 | **0.62** | 0.44 | **0.75** |
| | M-Entropy | 0.02 | **0.06** | 0.09 | **0.17** | 0.17 | **0.30** | 0.23 | **0.46** | 0.31 | **0.62** |
| **LLaMA-7B** | Perplexity | 0.00 | **0.02** | 0.02 | **0.11** | 0.04 | **0.28** | 0.08 | **0.47** | 0.15 | **0.65** |
| | Zlib | 0.01 | **0.02** | 0.03 | **0.08** | 0.05 | **0.23** | 0.08 | **0.44** | 0.15 | **0.63** |
| | MIN-K% | 0.01 | **0.03** | 0.05 | **0.10** | 0.09 | **0.21** | 0.12 | **0.35** | 0.19 | **0.51** |
| | M-Entropy | 0.00 | **0.01** | 0.01 | **0.06** | 0.03 | **0.17** | 0.05 | **0.33** | 0.08 | **0.49** |
| **Pythia-6.9B** | Perplexity | 0.01 | **0.02** | 0.03 | **0.06** | 0.06 | **0.15** | 0.11 | **0.27** | 0.18 | **0.43** |
| | Zlib | 0.01 | **0.02** | 0.03 | **0.07** | 0.07 | **0.16** | 0.12 | **0.29** | 0.18 | **0.45** |
| | MIN-K% | 0.03 | **0.04** | 0.07 | **0.12** | 0.13 | **0.24** | 0.20 | **0.37** | 0.29 | **0.50** |
| | M-Entropy | 0.01 | **0.02** | 0.05 | **0.08** | 0.09 | **0.15** | 0.12 | **0.26** | 0.18 | **0.41** |

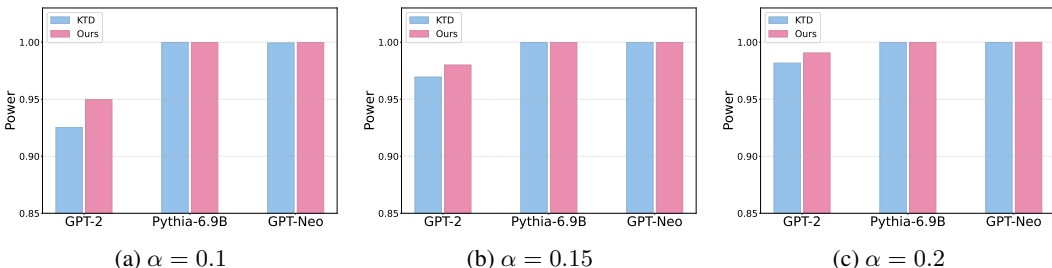

(a) $\alpha = 0.1$        (b) $\alpha = 0.15$        (c) $\alpha = 0.2$

Figure 3: Comparison of power on BBC Real Time with $\alpha = 0.1, 0.15, 0.2$.

comparably on the others. In summary, our method not only guarantees strict FDR control but also demonstrates superior power. More detail results are provided in Appendix F.8.

**Robustness of FDR Control to Variations in $\pi_{\text{test}}$.** We assess the robustness of our method by evaluating its performance varying the proportion of training members in the test data, $\pi_{\text{test}}$. This analysis utilizes the MiniGPT-4 vision-language model (Zhu et al., 2024a) and the VL-MIA/Flickr dataset (Li et al., 2024b). The detection score $T(X)$ is based on the MaxRényi-K% (Li et al., 2024b), configured with hyperparameters $K = 100$ and $\gamma = 0.5$ (see Equation (3)). The results presented in Figure 4 demonstrate that the achieved FDR is consistently bounded by the nominal level $\alpha$ across all tested values of $\pi_{\text{test}}$, thereby validating the effectiveness of our approach.

## 4.3 ABLATION STUDY

**Effect of the scaling procedure.** To examine the effectiveness of our scaled p-value in Line 4 of Algorithm 1, we compare with the vanilla method that directly uses the original p-value in Equation (7), which is close to Algorithm 2 in previous work (Jin & Candès, 2023). Table 1 presents that our method consistently achieves higher power than vanilla. For example, at the target FDR levels $\alpha = 0.1$, our method improves the power from 0.05 to 0.12 using MIN-K% score under NeoX-20B. We also provide the empirical FDR in Table 5, which demonstrates that the FDRs of our method are closer to the target level, thereby leading to superior powers. Further experimental results about ArXivTection are presented in Appendix F.3. More results under fine-tuning settings and those using reference-based scores are provided in Appendix F.6 and Appendix F.5, respectively.

**Impact of Calibration Set Size.** We investigate the effect of calibration set size on our method by evaluating FDR and power on ArXivTection with different models. Specifically, let $\rho = n/m$

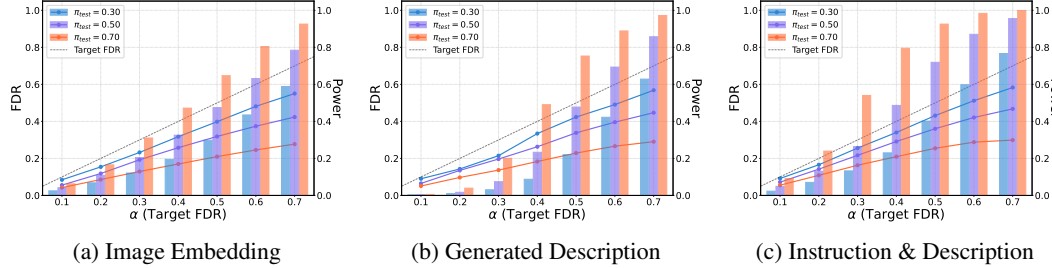

Figure 4: FDR (solid lines) and power (bars) achieved by our method on MiniGPT-4 with the VL-MIA/Flickr dataset, evaluated across various data usage proportions of the test set $\pi_{\text{test}}$ and target FDR levels $\alpha$. All results are based on the MaxRényi-K% score calculated from three different input components: (a) the image embedding, (b) the generated description, and (c) the instruction combined with the description.

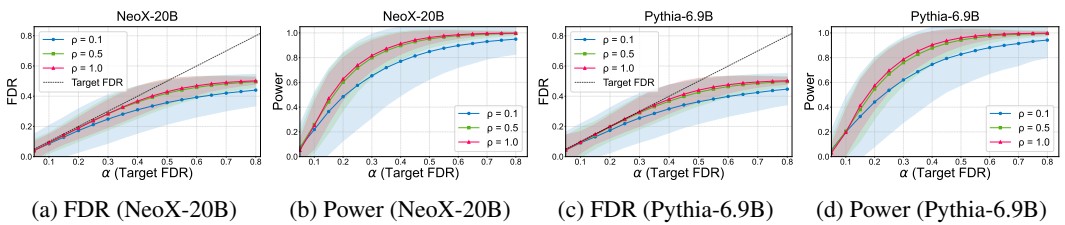

Figure 5: FDR curve achieved by our method under varying calibration set sizes. The parameter $\rho = n/m$ represents the ratio of the calibration set size ($n$) to the test set size ($m$). Shaded regions correspond to the mean ± one standard deviation.

denote the ratio of the calibration set size to the test set size. We vary $\rho$ in $\{0.1, 0.5, 1\}$. Figure 5 presents that our method effectively controls FDR across all tested $\rho$ values. In addition, increasing the calibration set size reduces the variance of both the False Discovery Proportion (FDP) and power, resulting in more stable training data identification.

**Robustness to the hyperparameter $\eta$.** As described in Section 3.2, our approach requires selecting a quantile threshold $\eta$ to construct the score region $\mathcal{R}$ from the calibration set. We test the sensitivity to this choice with GPT-NeoX-20B on ArXivTection. The results in Figure 8 demonstrate that our method robustly controls the FDR across all tested values of $\eta$, which aligns with the guarantee provided by Theorem 1. In practice, $\eta$ should be chosen to identify a region $\mathcal{R}$ sparsely populated by member data. This suggests selecting a small $\eta$ but one that is not so extreme as to cause instability in the estimation. In this paper, we set $\eta = 0.05$ by default.

## 5 DISCUSSION

**Improving power by adjusted moment estimator.** In some auditing scenarios, a calibration set containing a mix of confirmed **members** and **non-members** is available, though with an arbitrary membership proportion. Such a set can be constructed by sampling from canonical public datasets known to be part of the model's training corpus (e.g., the Pile (Gao et al., 2020)) or by identifying instances of verbatim memorization. We argue that the information from known members can significantly enhance power. Accordingly, we propose a corresponding method based on the method of moments to estimate the data usage proportion.

For convenience, we define $\pi_0 = 1 - \pi_{\text{test}}$. The raw estimator by moment for $\pi_0$ is:

$$\hat{\pi}_{0,\text{raw}} = \frac{\hat{\mu}_1 - \hat{\mu}_{\text{test}}}{\hat{\mu}_1 - \hat{\mu}_0}$$

where $\hat{\mu}_0$, $\hat{\mu}_1$, and $\hat{\mu}_{\text{test}}$ are means of the detection scores derived from the non-member calibration set $\mathcal{D}_{\text{cal}}^0$, member calibration set $\mathcal{D}_{\text{cal}}^1$, and $\mathcal{D}_{\text{test}}$, respectively. To mitigate the positive bias in $1/\hat{\pi}_{0,\text{raw}}$

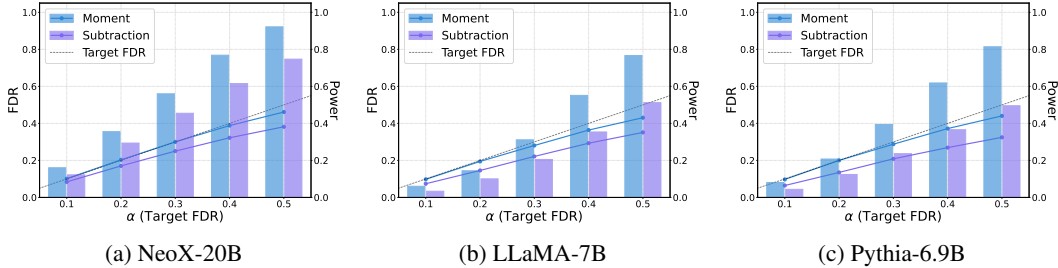

| (a) NeoX-20B | (b) LLaMA-7B | (c) Pythia-6.9B |

Figure 6: Performance of the subtraction and adjusted moment estimators on the WikiMIA dataset. Each plot shows the realized FDR (solid lines) and statistical power (bars) for a given model.

caused by Jensen's inequality, we introduce a bias-corrected estimator for the reciprocal $\theta_{1/\pi_0} = 1/\pi_0$, by subtracting an estimate of the leading bias term:

$$\hat{\theta}_{1/\pi_0} = \frac{1}{\hat{\pi}_{0,\text{raw}}} - \frac{\widehat{\text{Var}}(\hat{\pi}_{0,\text{raw}})}{\hat{\pi}_{0,\text{raw}}^3} \tag{13}$$

To implement this, we approximate the variance $\text{Var}(\hat{\pi}_{0,\text{raw}})$ using the Delta method:

$$\widehat{\text{Var}}(\hat{\pi}_{0,\text{raw}}) = \frac{1}{(\hat{\mu}_1 - \hat{\mu}_0)^2} \left[ \hat{\pi}_{0,\text{raw}}^2 \frac{\hat{\sigma}_0^2}{n_0} + (1 - \hat{\pi}_{0,\text{raw}})^2 \frac{\hat{\sigma}_1^2}{n_1} + \frac{\hat{\sigma}_{\text{test}}^2}{m} \right] \tag{14}$$

where $n_0 = |\mathcal{D}_{\text{cal}}^0|$, $n_1 = |\mathcal{D}_{\text{cal}}^1|$, and $m = |\mathcal{D}_{\text{test}}|$. The terms $\hat{\sigma}_0^2, \hat{\sigma}_1^2, \hat{\sigma}_{\text{test}}^2$ are the corresponding sample variances. The final estimator used in our algorithm is $\hat{\pi}_{\text{mom}} = 1 - 1/\hat{\theta}_{1/\pi_0}$.

The corresponding results in Figure 6 show that the adjusted moment estimator achieves higher power while maintaining FDR control. This is mainly done by running up the target FDR budget by a more precise estimate for $\pi_{\text{test}}$. We can establish the following proposition:

**Proposition 2.** *Assume the detection scores for member, non-member, and test distributions have finite first and second moments. As the sample sizes of the calibration and test sets $n_0, n_1, m \to \infty$, the estimator $\hat{\pi}_{mom}$ is a consistent estimator for $\pi_{test}$. That is:*

$$\hat{\pi}_{mom} \xrightarrow{p} \pi_{test},$$

*where $\xrightarrow{p}$ denotes convergence in probability.*

The corresponding proof is provided in the Appendix C.3. This proposition shows that the adjusted moment estimator converges to the $\pi_{\text{test}}$ as the data size is sufficiently large, thereby leading to higher power for training data identification. The details about $\hat{\pi}_{\text{mom}}$ is provided in Appendix D.

## 6 CONCLUSION

In this paper, we introduce Provable Training Data Identification (PTDI), a provable method that identifies training data with provable FDR control. Our approach leverages conformal p-values and the Benjamini-Hochberg procedure to achieve distribution-free guarantees under the practical assumption that only a set of non-training data is required for calibration. To significantly enhance power, we introduce a conservative estimator for the data usage proportion, which enables a p-value scaling technique that boosts the discovery number while maintaining theoretical rigor. For scenarios where auditors have access to some confirmed training data, we also propose an enhanced estimator to further improve performance. Extensive experiments demonstrate that PTDI consistently achieves higher power than prior methods with strict FDR control. In summary, our method provides a general and robust solution for discovering training data with strong theoretical guarantees, and it can be readily integrated with a wide range of existing detection scores across diverse settings.

**Limitations** Though our method provides rigorous theoretical guarantees, it requires a calibration set of unseen data that is distributionally similar to the test set. While obtaining such data (e.g., post-cutoff-date content) is feasible for well-defined domains such as benchmarks centered on math or copyright works for a certain painter, it is challenging for highly heterogeneous test data. In such scenarios, assembling a representative calibration set of unseen data is difficult. A significant distributional mismatch between the calibration data and the test data can invalidate the FDR guarantee.

## 7 REPRODUCIBILITY STATEMENT

We have made every effort to ensure that the results presented in this paper are reproducible. All code and datasets have been made publicly available in an anonymous repository to facilitate replication and verification. We have provided a full description of our method in Algorithm 1, to assist others in reproducing our experiments.

Additionally, the text and image dataset used in our paper, such as WikiMIA and VL-MIA/Flickr, are publicly available, ensuring consistent and reproducible evaluation results. We believe these measures will enable other researchers to reproduce our work and further advance the field.

## 8 THE USE OF LARGE LANGUAGE MODELS

In this paper, we only employ large language models to refine the clarity and readability of the manuscript. It is important to note that the LLM was not involved in the ideation, research methodology, or experimental design. All research concepts, ideas, and analyses were independently developed and carried out by the authors. The contributions of the LLM were solely focused on improving the linguistic quality of the paper, with no involvement in the scientific content or data analysis.

The authors take full responsibility for the content of the manuscript, including any text generated or polished by the LLM. We have ensured that the LLM-generated text adheres to ethical guidelines and does not contribute to plagiarism or scientific misconduct.

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

# A  RELATED WORK

**Training Data Detection in Large-Scale Models.**  Identifying training data within large-scale models is a critical task with significant real-world implications, including ensuring fair model evaluation and providing credible evidence in copyright litigation. A primary concern in academic research is data contamination, where benchmark data leaks into the training set, leading to untrustworthy evaluation results (Magar & Schwartz, 2022; Zhou et al., 2023). To address these issues, numerous studies have developed heuristic detection scores (Mattern et al., 2023; Xie et al., 2024; Zhang et al., 2024; Raoof et al., 2025). These include metrics like perplexity (Carlini et al., 2022), MIN-k% (Shi et al., 2024), MIN-k%++ (Zhang et al., 2025c) for LLMs, and MaxRényi-K% for VLMs (Li et al., 2024b). However, these methods treat the task as a binary classification problem for individual points and lack the theoretical guarantees.

Seeking to add statistical rigor, another line of work provides theoretical guarantees. For instance, Dekoninck et al. (2024) uses multiple reference models to construct valid statistical tests, and Oren et al. (2023) leverages exchangeability for statistical inference. A key limitation, however, is that their guarantees apply only to dataset-level hypotheses—for example, determining if an entire dataset as a whole is contaminated. They are not designed for the fine-grained task of selecting a credible subset of individual data points. This is insufficient for practical applications, such as a copyright holder providing a specific list of infringed works, or an evaluator removing specific contaminated examples from a benchmark, a function supported by toolkits like lm-evaluation-harness (Gao et al., 2024). In this paper, we provide a method that determines a subset from a given dataset with a strict FDR control guarantee.

**Membership Inference Attacks.**  From a privacy perspective, Membership Inference Attacks (MIAs) aim to determine if a specific data point was used to train a target model, which could expose sensitive information (Shokri et al., 2017; Yeom et al., 2018; Salem et al., 2019). A significant body of work treats this as a binary classification problem, relying on scores computed without reference models, such as loss (Yeom et al., 2018), entropy (Yeom et al., 2018), confidence (Liu et al., 2019) and gradient norm Nasr et al. (2019); Sablayrolles et al. (2019). While accessible, this approach focuses on average classification accuracy. Shifting to a more statistically-minded viewpoint, many prominent works correctly frame MIA as a hypothesis test, including Attack-P (Ye et al., 2022) and other prominent works (Carlini et al., 2022; Zarifzadeh et al., 2024). These approaches prioritize metrics like the true positive rate (TPR) at a low false positive rate (FPR), but this still relies on an average-case error metric and fails to provide a formal statistical guarantee for any individual inference. Furthermore, some attacks like Attack-R (Ye et al., 2022) and BMIA (Liu et al., 2025) estimate the conditional distribution of individual data points to control type-I error for specific inferences, but they require at least one reference model, making them unsuitable for detecting pre-training data in LLMs. Moreover, the average type-I error (FPR) is ill-suited for multiple-hypothesis testing, where controlling the FDR is more appropriate for ensuring credible evidence across the selected membership set. In this work, we propose a versatile method that ensures strict FDR control and integrates seamlessly with most MIA methods.

**False Discovery Rate Control in Training Data Identification.**  Recognizing the limitations of binary classification, recent work has shifted towards controlling the False Discovery Rate (FDR), which is a more appropriate error metric when the goal is to produce a credible set of training data. The Benjamini-Hochberg (BH) procedure (Benjamini & Hochberg, 1995; Benjamini & Yekutieli, 2001) is a standard tool for this, but it is conservative since it assumes that all null hypotheses might be true (number of true nulls $m_0 = m$). Thus, Storey (2002) proposed to estimate $\pi_0$ (null hypotheses proportion) using the independent p-values, which yields a less stringent procedure and significantly increases power. However, its application to modern machine learning is challenging due to the difficulty of constructing valid p-values under realistic assumptions. To address this, recent work Bates et al. (2023); Jin & Candès (2023) introduced the use of conformal p-values (Vovk et al., 2003; 2005) to guarantee FDR control. Notably, Bates et al. (2023) formally proved that Storey's correction remains valid for conformal p-values despite their positive dependence (PRDS).

Regarding the specific task of training data detection, Hu et al. (2025) proposed a method based on knockoff statistics to control FDR. However, its effectiveness hinges on generating high-quality

knockoffs (distributed symmetrically), a challenging task that can lead to unstable FDR control in practice. Our work builds on the conformal inference framework but introduces score-based estimators. By leveraging the distributional properties of detection scores, our method provides a versatile solution that adapts to *different auditing scenarios* (e.g., constructing specific estimators depending on whether known members are available), thereby achieving strictly higher power while maintaining rigorous FDR control.

**Estimating Data Usage Proportion.**    A key component of our method's ability to improve power is the estimation of the data usage proportion. This problem was recently formalized as Dataset Usage Cardinality Inference (DUCI) by Tong et al. (2025). Their approach, however, requires training reference models to estimate the necessary statistics, rendering it unsuitable for pre-training data detection in large-scale models where the training process is opaque and prohibitively expensive. In contrast, our proposed subtraction estimator is significantly more practical, as it only requires access to the target model, a set of confirmed non-member data, and the test set, making it a more versatile tool for real-world auditing scenarios.

## B    TABLE OF NOTATIONS

Table 2: Summary of notations used in the paper.

| Notation | Description |
| --- | --- |
| $\theta$ | The target model. |
| $\mathcal{D}_{\text{train}}$ | The set of data used to train the model $\theta$. |
| $X, X_i$ | A data point and the $i$-th data point, respectively. |
| $M, M_i$ | true membership label for a data point $X$, $X_i$ (1 if it is in traning set $\mathcal{D}_{\text{train}}$, otherwise 0). |
| $\mathcal{D}_{\text{cal}}, \mathcal{D}_{\text{test}}$ | The calibration set and the test set, respectively. |
| $n, m$ | The number of samples in the calibration and test sets, respectively. |
| $T(X; \theta)$ | The detection score for data point $X$ (e.g., perplexity). Lower is more member-like. |
| $H_j$ | The null hypothesis that test point $X_{n+j}$ is a non-member ($M_{n+j} = 0$). |
| $\mathcal{S}$ | The selected subset of indices from $\mathcal{D}_{\text{test}}$ rejected as null hypotheses. |
| FDR | The expected proportion of false discoveries in $\mathcal{S}$. |
| Power | The expected proportion of all true members that are correctly identified in $\mathcal{S}$. |
| $\alpha$ | The target False Discovery Rate (FDR) level. |
| $\pi_{\text{test}}$ | The true proportion of training members in the test set. |
| $\hat{\pi}_{\text{test}}$ | An estimate of the data usage proportion $\pi_{\text{test}}$. |
| $p_j$ | The initial conformal p-value for the $j$-th test point. |
| $\tilde{p}_j$ | The scaled p-value, calculated as $(1 - \hat{\pi}_{\text{test}})p_j$. |
| $\mathcal{E}$ | A generic estimator function for the data usage proportion. |
| $\hat{\pi}_{\text{sub}}$ | The subtraction-based estimator for $\pi_{\text{test}}$. |
| $\hat{\pi}_{\text{mom}}$ | The adjusted moment-based estimator for $\pi_{\text{test}}$. |
| $\mathcal{R}$ | A region of scores used by the subtraction estimator. |
| $\eta$ | A quantile hyperparameter used to define the region $\mathcal{R}$. |
| $\mu_0, \mu_1$ | The true mean of detection scores for non-members and members. |
| $\hat{\mu}_0, \hat{\mu}_1$ | The sample mean of detection scores for non-members and members. |
| $m_0, m_1$ | The number of non-member and member samples in the test set, respectively. |

## C  PROOFS

### C.1  THE PROOF OF PROPOSITION 1

*Proof.* The expectation is taken over the randomness of both the calibration set $\mathcal{D}_{\text{cal}}$ (which determines the random region $\mathcal{R}$) and the test set. We use the law of total expectation by first conditioning on a fixed region $\mathcal{R}$.

Note that $\mathcal{R}$ is chosen such that $\frac{1}{n}\sum_{i=1}^{n}\mathbb{1}\{T(X_i)\in\mathcal{R}\}=\eta$. Let $K=\sum_{j=1}^{m}\mathbb{1}\{T(X_{n+j})\in\mathcal{R}\}$ be the random count of test points in the region $\mathcal{R}$. Then we have:

$$1-\hat{\pi}_{\text{sub}}=\frac{K+1}{(m+1)\eta}$$

The expression inside the expectation is therefore:

$$\frac{1-\pi_{\text{test}}}{1-\hat{\pi}_{\text{sub}}}=\frac{\eta(m+1)(1-\pi_{\text{test}})}{K+1}$$

Now, we take the conditional expectation with respect to the test set, for a fixed region $\mathcal{R}$:

$$\mathbb{E}_{K|\mathcal{R}}\left[\frac{1-\pi_{\text{test}}}{1-\hat{\pi}_{\text{sub}}}\right]=\eta(m+1)(1-\pi_{\text{test}})\cdot\mathbb{E}_{K|\mathcal{R}}\left[\frac{1}{K+1}\right]$$

The count $K$ follows a binomial distribution $K\sim\text{Bin}(m,p_K)$, where $p_K=\int_{\mathcal{R}}p(t_{\text{test}})dt$. A key property of the binomial distribution is the exact closed-form solution for the expectation of $1/(K+1)$:

$$\mathbb{E}_{K|\mathcal{R}}\left[\frac{1}{K+1}\right]=\frac{1-(1-p_K)^{m+1}}{(m+1)p_K}$$

Substituting this back into our expression for the conditional expectation gives:

$$\mathbb{E}_{K|\mathcal{R}}\left[\frac{1-\pi_{\text{test}}}{1-\hat{\pi}_{\text{sub}}}\right]=\eta(m+1)(1-\pi_{\text{test}})\left(\frac{1-(1-p_K)^{m+1}}{(m+1)p_K}\right)$$

$$=\frac{\eta(1-\pi_{\text{test}})}{p_K}\left(1-(1-p_K)^{m+1}\right)$$

From the initial problem setup, we have the inequality $p_K\geqslant(1-\pi_{\text{test}})\eta$, which implies $\frac{\eta(1-\pi_{\text{test}})}{p_K}\leqslant 1$. Furthermore, since $p_K\in[0,1]$ and $m\geqslant 0$, the term $(1-(1-p_K)^{m+1})$ is also less than or equal to 1. The product of two non-negative numbers that are both less than or equal to 1 must also be less than or equal to 1. Thus, for any fixed region $\mathcal{R}$, the conditional expectation is bounded:

$$\mathbb{E}_{K|\mathcal{R}}\left[\frac{1-\pi_{\text{test}}}{1-\hat{\pi}_{\text{sub}}}\right]\leqslant 1$$

Since this inequality holds for any region $\mathcal{R}$ that could be realized from the calibration set, the total expectation over both the calibration and test sets must also be bounded by 1:

$$\mathbb{E}\left[\frac{1-\pi_{\text{test}}}{1-\hat{\pi}_{\text{sub}}}\right]=\mathbb{E}_{\mathcal{R}}\left[\mathbb{E}_{K|\mathcal{R}}\left[\frac{1-\pi_{\text{test}}}{1-\hat{\pi}_{\text{sub}}}\right]\right]\leqslant\mathbb{E}_{\mathcal{R}}[1]=1$$

This completes the proof.  □

### C.2  THE PROOF OF THEOREM 1

**Lemma 2** (Classical FDR Control under PRDS (Benjamini & Yekutieli, 2001))**.** *Given a set of p-values $\{p_j\}_{j=1}^{m}$ from $m$ hypothesis tests, of which $m_0$ are true null hypotheses. If the p-values corresponding to the true nulls are valid (i.e., they are super-uniformly distributed under the null), and if the entire p-value vector satisfies the Positive Regression Dependency on a Subset (PRDS) property, then the standard Benjamini-Hochberg (BH) procedure at a target level $\alpha$ controls the False Discovery Rate (FDR) such that:*

$$\text{FDR}\leqslant\frac{m_0}{m}\alpha.$$

**Proposition 3** (FDR Control for the BH Procedure on $p_j$). *Let the p-values $\{p_j\}_{j=1}^{m}$ be constructed as defined in Equation* (7)*. Assume that each sample's membership status in the test set $\mathcal{D}_{test}$ is an independent Bernoulli trial, where the probability of being a member ($M = 1$) is $\pi_{test}$. Further assume that the p-value vector $(p_1, \ldots, p_m)$ satisfies the PRDS property.*

*If the standard Benjamini-Hochberg (BH) procedure is applied directly to these p-values $\{p_j^0\}$ at a target level $\alpha$, then its FDR is controlled such that:*

$$\mathrm{FDR} \leqslant \alpha \cdot (1 - \pi_{test}).$$

*Proof.* First, we verify that the p-values $\{p_j\}$ generated by Equation (7) satisfy the preconditions of Lemma 2.

For any true null hypothesis $H_j$ (i.e., $M_{n+j} = 0$), the p-value $p_j$ is constructed using a calibration set composed entirely of non-members. According to the principles of conformal prediction, when the test sample $X_{n+j}$ is also a non-member, its score $T_{n+j}$ is exchangeable with the scores from the calibration set. This construction guarantees that $p_j$ is super-uniformly distributed under the null hypothesis, meaning $P(p_j \leqslant t \mid j \in \mathcal{H}_0) \leqslant t$ for all $t \in [0, 1]$.

The PRDS property is validated by Lemma B.1 of Jin & Candès (2023). Since both conditions are met, we can apply the conclusion of Lemma 2 for a fixed $\mathcal{D}_{\text{test}}$:

$$\mathrm{FDR} = \mathbb{E}[\mathrm{FDP} \mid \mathcal{D}_{\text{test}}] \leqslant \frac{m_0}{m}\alpha$$

Recall that $\pi_{\text{test}} = \Pr(M_j = 1)$. For a fixed test set $\mathcal{D}_{\text{test}}$, this probability corresponds to the actual proportion of members, i.e., $\pi_{\text{test}} = 1 - m_0/m$. By substituting, we obtain:

$$\mathrm{FDR} \leqslant (1 - \pi_{\text{test}}) \cdot \alpha.$$

$\square$

With these results, we are ready to prove the main theorem.

***Proof of Theorem 1.*** Let $\mathcal{S}$ be the rejection set from Algorithm 1. The FDR is defined as $\mathrm{FDR} = \mathbb{E}[\mathrm{FDP}]$. We analyze this expectation by conditioning on the random estimator $\hat{\pi}_{\text{sub}}$. By the law of total expectation:

$$\mathrm{FDR} = \mathbb{E}_{\hat{\pi}_{\text{sub}}}\left[\mathbb{E}\left[\mathrm{FDP}\big|\hat{\pi}_{\text{sub}}\right]\right]$$

For a fixed value of the estimator $\hat{\pi}_{\text{sub}}$, the procedure's rejection rule on the adjusted p-values $\{\tilde{p}_j\}$ is equivalent to applying the standard BH procedure to the original p-values $\{p_j\}$ at a modified, random level of $\alpha' = \alpha/(1 - \hat{\pi}_{\text{sub}})$.

We can now apply Proposition 3 to analyze the inner conditional expectation. The proportion's unconditional form states that a standard BH procedure at level $\alpha'$ provides an FDR guarantee of $\pi_0 \alpha'$. Therefore, the conditional FDR of our procedure is bounded as:

$$\mathbb{E}\left[\mathrm{FDP}\big|\hat{\pi}_{\text{sub}}\right] \leqslant (1 - \pi_{\text{sub}}) \cdot \alpha' = \alpha \cdot \frac{1 - \pi_{\text{sub}}}{1 - \hat{\pi}_{\text{sub}}}$$

Finally, we take the outer expectation with respect to the randomness in $\hat{\pi}_0$ and apply the Proposition 1:

$$\mathrm{FDR} \leqslant \mathbb{E}_{\hat{\pi}_{\text{sub}}}\left[\alpha \cdot \frac{1 - \pi_{\text{sub}}}{1 - \hat{\pi}_{\text{sub}}}\right] = \alpha \cdot \mathbb{E}\left[\frac{1 - \pi_{\text{test}}}{1 - \hat{\pi}_{\text{sub}}}\right] \leqslant \alpha \qquad (15)$$

This completes the proof. $\square$

## C.3 THE PROOF OF PROPOSITION 2

*Proof.* We prove the consistency of $\hat{\pi}_{\text{mom}}$ by relying on the Weak Law of Large Numbers (WLLN) and the Continuous Mapping Theorem (CMT). Let $\mu_0, \mu_1$ and $\sigma_0^2, \sigma_1^2$ be the true means and variances of the detection scores for the non-member and member populations, respectively. The true mean of the test set is $\mu_{\text{test}} = (1 - \pi_{\text{test}})\mu_0 + \pi_{\text{test}}\mu_1$, and we define the true non-member proportion as $\pi_0 = 1 - \pi_{\text{test}}$.

Under the assumption of finite moments, the WLLN ensures that the sample means $\hat{\mu}_0, \hat{\mu}_1, \hat{\mu}_\text{test}$ and sample variances $\hat{\sigma}_0^2, \hat{\sigma}_1^2, \hat{\sigma}_\text{test}^2$ converge in probability to their true population counterparts. Since the raw estimator $\hat{\pi}_{0,\text{raw}}$ is a continuous function of these sample means, the CMT implies its convergence in probability. Specifically, assuming $\mu_1 \neq \mu_0$ for the score to be informative, we have

$$\hat{\pi}_{0,\text{raw}} = \frac{\hat{\mu}_1 - \hat{\mu}_\text{test}}{\hat{\mu}_1 - \hat{\mu}_0} \xrightarrow{p} \frac{\mu_1 - \mu_\text{test}}{\mu_1 - \mu_0} = \frac{\mu_1 - ((1 - \pi_\text{test})\mu_0 + \pi_\text{test}\mu_1)}{\mu_1 - \mu_0} = \pi_0.$$

Next, we consider the bias-correction term $\frac{\widehat{\text{Var}}(\hat{\pi}_{0,\text{raw}})}{\hat{\pi}_{0,\text{raw}}^3}$. Recall that

$$\widehat{\text{Var}}(\hat{\pi}_{0,\text{raw}}) = \frac{1}{(\hat{\mu}_1 - \hat{\mu}_0)^2} \left[ \hat{\pi}_{0,\text{raw}}^2 \frac{\hat{\sigma}_0^2}{n_0} + (1 - \hat{\pi}_{0,\text{raw}})^2 \frac{\hat{\sigma}_1^2}{n_1} + \frac{\hat{\sigma}_\text{test}^2}{m} \right],$$

where $n_0 = |\mathcal{D}_\text{cal}^0|$, $n_1 = |\mathcal{D}_\text{cal}^1|$, and $m = |\mathcal{D}_\text{test}|$. As $n_0, n_1, m \to \infty$, the estimated variance $\widehat{\text{Var}}(\hat{\pi}_{0,\text{raw}})$ converges in probability to zero because its constituent terms are scaled by $1/n_0, 1/n_1$, or $1/m$, and the $\hat{\pi}_{0,\text{raw}}^3$ converges in probability to the non-zero constant $\pi_0^3$. Thus, the entire correction term converges to zero by Slutsky's theorem. It follows that $\hat{\theta}_{1/\pi_0} = 1/\hat{\pi}_{0,\text{raw}} - \frac{\widehat{\text{Var}}(\hat{\pi}_{0,\text{raw}})}{\hat{\pi}_{0,\text{raw}}^3} \xrightarrow{p} 1/\pi_0$. Finally, since the final estimator is a continuous transformation of $\hat{\theta}_{1/\pi_0}$, another application of the CMT yields the desired result:

$$\hat{\pi}_\text{mom} = 1 - \frac{1}{\hat{\theta}_{1/\pi_0}} \xrightarrow{p} 1 - \frac{1}{1/\pi_0} = 1 - \pi_0 = \pi_\text{test}.$$

This completes the proof. $\qquad\square$

## C.4    SCALING PROCEDURE IMPROVES POWER

**Theorem 3.** *Let $\mathcal{S}_0$ and $\mathcal{S}_1$ be the selection sets from the standard Benjamini-Hochberg procedure and the scaling procedure, respectively. If $0 < \hat{\pi}_{test} < 1$, then we have:*

$$Power(\mathcal{S}_1) \geqslant Power(\mathcal{S}_0)$$

*Proof.* Let $p_{(1)} \leqslant \cdots \leqslant p_{(m)}$ be the sorted p-values. Suppose that selection sets are determined by the stopping indices $k_0$ and $k_1$:

$$\mathcal{S}_0 = \{j \mid p_j \leqslant p_{(k_0)}\}, \ \mathcal{S}_1 = \{j \mid p_j \leqslant p_{(k_1)}\},$$

where $k_0 = \max\left\{k \mid p_{(k)} \leqslant \frac{k}{m}\alpha\right\}$ and $k_1 = \max\left\{k \mid \tilde{p}_{(k)} \leqslant \frac{k}{m}\alpha\right\}$.

Substituting $\tilde{p}_{(k)} = (1 - \hat{\pi}_\text{test})p_{(k)}$ into the condition for $k_1$ yields the equivalent inequality:

$$\tilde{p}_{(k)} \leqslant \frac{k}{m}\alpha \iff (1 - \hat{\pi}_\text{test})p_{(k)} \leqslant \frac{k}{m}\alpha \iff p_{(k)} \leqslant \frac{k}{m}\left(\frac{\alpha}{1 - \hat{\pi}_\text{test}}\right)$$

Since $0 < \hat{\pi}_\text{test} < 1$, strictly weaker rejection criterion applies to $\mathcal{S}_1$:

$$p_{(k)} \leqslant \frac{k}{m}\alpha \implies p_{(k)} \leqslant \frac{k}{m}\alpha \leqslant \frac{k}{m}\left(\frac{\alpha}{1 - \hat{\pi}_\text{test}}\right)$$

This implication ensures that any index $k$ satisfying the condition for $\mathcal{S}_0$ also satisfies it for $\mathcal{S}_1$. Therefore:

$$k_0 \leqslant k_1 \implies p_{(k_0)} \leqslant p_{(k_1)} \implies \mathcal{S}_0 \subseteq \mathcal{S}_1$$

By the definition of power, we have:

$$Power(\mathcal{S}_1) \geqslant Power(\mathcal{S}_0) \tag{16}$$

This completes the proof. $\qquad\square$

# D   DETAIL DERIVATION FOR ADJUSTED MOMENT ESTIMATOR

We first define the raw moment estimator, $\hat{\pi}_{0,\text{raw}}$, as follows:

$$\hat{\pi}_{0,\text{raw}} = \frac{\hat{\mu}_1 - \hat{\mu}_{\text{test}}}{\hat{\mu}_1 - \hat{\mu}_0}, \tag{17}$$

where $\hat{\mu}_0, \hat{\mu}_1$, and $\hat{\mu}_{\text{test}}$ are the sample means from their respective datasets. By the Law of Large Numbers and the Continuous Mapping Theorem, $\hat{\pi}_{0,\text{raw}}$ is a consistent estimator of the true proportion $\pi_0$, i.e., $\hat{\pi}_{0,\text{raw}} \xrightarrow{p} \pi_0$.

However, our interest lies in the expectation of its reciprocal, $E[1/\hat{\pi}_{0,\text{raw}}]$. Since the function $f(x) = 1/x$ is convex, by Jensen's inequality, we have $E[1/\hat{\pi}_{0,\text{raw}}] > 1/E[\hat{\pi}_{0,\text{raw}}]$, which indicates that $1/\hat{\pi}_{0,\text{raw}}$ is a biased estimator of $1/\pi_0$, and it is likely to obtain an undesirable property such that $E[1/\hat{\pi}_{0,\text{raw}}] > 1/\hat{\pi}_{0,\text{raw}}$ (see the Equation (15), which requires the oppsite property).

To quantify this bias, we perform a second-order Taylor expansion of the function $f(\hat{\pi}_{0,\text{raw}}) = 1/\hat{\pi}_{0,\text{raw}}$ around the true value $\pi_0$:

$$\frac{1}{\hat{\pi}_{0,\text{raw}}} \approx \frac{1}{\pi_0} - \frac{1}{\pi_0^2}(\hat{\pi}_{0,\text{raw}} - \pi_0) + \frac{1}{\pi_0^3}(\hat{\pi}_{0,\text{raw}} - \pi_0)^2.$$

Taking the expectation of both sides, we get:

$$E\left[\frac{1}{\hat{\pi}_{0,\text{raw}}}\right] \approx E\left[\frac{1}{\pi_0}\right] - \frac{1}{\pi_0^2}E[\hat{\pi}_{0,\text{raw}} - \pi_0] + \frac{1}{\pi_0^3}E[(\hat{\pi}_{0,\text{raw}} - \pi_0)^2].$$

Since $\hat{\pi}_{0,\text{raw}}$ is asymptotically unbiased, $E[\hat{\pi}_{0,\text{raw}} - \pi_0] \approx 0$. By the definition of variance, $E[(\hat{\pi}_{0,\text{raw}} - \pi_0)^2] \approx \text{Var}(\hat{\pi}_{0,\text{raw}})$. Thus, the equation simplifies to:

$$E\left[\frac{1}{\hat{\pi}_{0,\text{raw}}}\right] \approx \frac{1}{\pi_0} + \frac{\text{Var}(\hat{\pi}_{0,\text{raw}})}{\pi_0^3}.$$

The term $\text{Var}(\hat{\pi}_{0,\text{raw}})/\pi_0^3$ is the leading source of bias. It is a positive value, indicating that the naive estimator usually overestimates $1/\pi_0$. We define $g(\mu_0, \mu_1, \mu_{\text{test}}) = \frac{\mu_1 - \mu_{\text{test}}}{\mu_1 - \mu_0}$. According to the first-order Delta method, the variance of $\hat{\pi}_{0,\text{raw}}$ can be approximated as:

$$\text{Var}(\hat{\pi}_{raw}) \approx \left(\frac{\partial g}{\partial \mu_0}\right)^2 \text{Var}(\hat{\mu}_0) + \left(\frac{\partial g}{\partial \mu_1}\right)^2 \text{Var}(\hat{\mu}_1) + \left(\frac{\partial g}{\partial \mu_{\text{test}}}\right)^2 \text{Var}$$

$$= \frac{1}{(\mu_1 - \mu_0)^2}\left[\pi_0^2 \frac{\sigma_0^2}{n_0} + (1 - \pi_0)^2 \frac{\sigma_1^2}{n_1} + \frac{\sigma_{\text{test}}^2}{m}\right],$$

where $\sigma_i^2$ is the population variance and $n_i$ is the sample size.

Finally, to obtain the estimator for the variance, $\widehat{\text{Var}}(\hat{\pi}_{0,\text{raw}})$, we replace all unknown population parameters with their corresponding sample estimates:

$$\widehat{\text{Var}}(\hat{\pi}_{0,\text{raw}}) = \frac{1}{(\hat{\mu}_1 - \hat{\mu}_0)^2}\left[\hat{\pi}_{0,\text{raw}}^2 \frac{\hat{\sigma}_0^2}{n_0} + (1 - \hat{\pi}_{0,\text{raw}})^2 \frac{\hat{\sigma}_1^2}{n_1} + \frac{\hat{\sigma}_{\text{test}}^2}{m}\right]$$

Thus, we construct a corrected estimator, $\hat{\theta}_{1/\pi_0}$, which is designed to subtract the estimated leading bias term:

$$\hat{\theta}_{1/\pi_0} = \frac{1}{\hat{\pi}_{0,\text{raw}}} - \frac{\widehat{\text{Var}}(\hat{\pi}_{0,\text{raw}})}{\hat{\pi}_{0,\text{raw}}^3} \tag{18}$$

From the proof of Theorem 1, we have the bound $\text{FDR} \leqslant \alpha \cdot \mathbb{E}\left[\frac{1 - \pi_{\text{test}}}{1 - \hat{\pi}_{\text{sub}}}\right]$. Let $\pi_0 = 1 - \pi_{\text{test}}$ be the true proportion of non-members and $\hat{\pi}_0 = 1 - \hat{\pi}_{\text{mom}}$ be our final estimate. The inequality can be rewritten as $\text{FDR} \leqslant \alpha \cdot \mathbb{E}\left[\frac{\pi_0}{\hat{\pi}_0}\right]$.

As established in Proposition 2, our estimator is consistent, meaning $\hat{\pi}_{\text{mom}} \xrightarrow{p} \pi_{\text{test}}$ as the sample sizes grow. This implies that $\hat{\pi}_0 \xrightarrow{p} \pi_0$, and by the continuous mapping theorem, the ratio $\frac{\pi_0}{\hat{\pi}_0}$ converges in probability to 1. This ensures that the FDR is controlled asymptotically.

**Theorem 4** (Asymptotic FDR Control). *Under the conditions of Proposition 2, the PTDI procedure using the adjusted moment estimator $\hat{\pi}_{mom}$ achieves asymptotic FDR control. Specifically,:*

$$\limsup_{n,m\to\infty} \text{FDR} \leqslant \alpha.$$

*Proof.* Recall from the proof of Theorem 1 (specifically Equation (15)) that the exact FDR is bounded by the expectation of the ratio between the true and estimated non-member proportions:

$$\text{FDR} \leqslant \alpha \cdot \mathbb{E}\left[\frac{1-\pi_{\text{test}}}{1-\hat{\pi}_{\text{mom}}}\right]. \tag{19}$$

By Proposition 2, we have $\hat{\pi}_{\text{mom}} \xrightarrow{p} \pi_{\text{test}}$. Let $g(x) = \frac{1-\pi_{\text{test}}}{1-x}$. Since $g(x)$ is continuous at $x = \pi_{\text{test}}$ (assuming $\pi_{\text{test}} < 1$), the Continuous Mapping Theorem implies:

$$\frac{1-\pi_{\text{test}}}{1-\hat{\pi}_{\text{mom}}} \xrightarrow{p} \frac{1-\pi_{\text{test}}}{1-\pi_{\text{test}}} = 1.$$

Assuming the estimator is bounded such that the ratio is uniformly integrable (which holds in practice as we clip $\hat{\pi}_{\text{mom}} \in [0, 1-\epsilon]$), convergence in probability implies convergence in mean. Taking the limit of Equation (19):

$$\limsup_{n,m\to\infty} \text{FDR} \leqslant \alpha \cdot \lim_{n,m\to\infty} \mathbb{E}\left[\frac{1-\pi_{\text{test}}}{1-\hat{\pi}_{\text{mom}}}\right] = \alpha \cdot 1 = \alpha.$$

This completes the proof. □

More critically for practical applications, the bias correction helps ensure control in finite samples. As derived previously, the term $1/\hat{\pi}_0$ in our method is calculated as

$$\frac{1}{1-\hat{\pi}_{\text{mom}}} = \hat{\theta}_{1/\pi_0} = \frac{1}{\hat{\pi}_{0,\text{raw}}} - \frac{\widehat{\text{Var}}(\hat{\pi}_{0,\text{raw}})}{\hat{\pi}_{0,\text{raw}}^3}.$$

Since the naive estimator $1/\hat{\pi}_{0,\text{raw}}$ overestimates $1/\pi_0$ on average due to Jensen's inequality, we subtract a positive term to correct for this bias. This correction makes our final estimator $\hat{\theta}_{1/\pi_0}$ conservative, such that its expectation is driven to be less than or equal to the true value $1/\pi_0$. This conservatism is precisely what leads to

$$\mathbb{E}\left[\frac{\pi_0}{\hat{\pi}_0}\right] = \pi_0 \mathbb{E}\left[\frac{1}{\hat{\pi}_0}\right] \lesssim \pi_0 \cdot \frac{1}{\pi_0} = 1,$$

thereby achieving valid FDR control in practice.

# E   EXPERIMENTAL DETAILS

Recall from Equation (4) that the false discovery rate (FDR) is defined as the expectation of the false discovery proportion,

$$\text{FDP} := \frac{\sum_{j=1}^{m} \mathbb{1}\{M_{n+j} = 0, j \in \mathcal{S}\}}{\max(|\mathcal{S}|, 1)}.$$

In our experiments, we report the empirical FDR by averaging FDP over 1000 times of Algorithm 1.

For the KTD comparison experiments shown in Figures 2, 3 and Appendix F.8, we follow the exact configuration of Hu et al. (2025) and use the full WikiMIA dataset (denoted "Total WikiMIA" in Table 3). For other experiments conducted on the WikiMIA benchmark, input text sequences were uniformly processed to a fixed length of 32 tokens. As for the experiment on VLMs, the generated token length is set to be 32.

Table 3: The test set and calibration set used in the main experiments.

| Dataset | Type | Member | Non-member | Total |
|---|---|---|---|---|
| WikiMIA | Test set | 193 | 194 | 387 |
| | Calibration set | / | 194 | 194 |
| | Original dataset | 387 | 388 | 775 |
| ArXivTection | Test set | 381 | 393 | 774 |
| | Calibration set | / | 393 | 393 |
| | Original dataset | 762 | 786 | 1,548 |
| VL-MIA/Flickr | Test set | 150 | 150 | 300 |
| | Calibration set | / | 150 | 150 |
| | Original dataset | 300 | 300 | 600 |
| VL-MIA/DALL-E | Test set | 148 | 148 | 296 |
| | Calibration set | / | 148 | 148 |
| | Original dataset | 296 | 296 | 592 |
| Total WikiMIA | Test set | 460 | 398 | 858 |
| | Calibration set | / | 398 | 398 |
| | Original Dataset | 861 | 789 | 1,650 |
| XSum | Test set | 2,790 | 2,875 | 5,665 |
| | Calibration set | / | 2,875 | 2,875 |
| | Original dataset | 5,581 | 5,751 | 11,332 |
| BBC Real Time | Test set | 1,638 | 1,674 | 3,312 |
| | Calibration set | / | 1,674 | 1,674 |
| | Original dataset | 3,277 | 3,349 | 6,626 |

### E.1 DATA SPLIT SETUP

Unless otherwise specified (e.g., when varying $\pi_{\text{test}}$), in each trial we randomly split the dataset into two equal halves. All non-members from one half are used to construct the calibration set $\mathcal{D}_{\text{cal}}$, while the other half serves as the test set $\mathcal{D}_{\text{test}}$. The detailed information of the constructed dataset is presented in Table 3.

For the experiment in Figure 4, in each trial, we further subsample the test set according to $\pi_{\text{test}}$. Specifically, we select $\pi_{\text{test}} \cdot |\mathcal{D}_{\text{test}}^0|$ points from $\mathcal{D}_{\text{test}}^1$ and $(1 - \pi_{\text{test}}) \cdot |\mathcal{D}_{\text{test}}^0|$ points from $\mathcal{D}_{\text{test}}^0$, where $\mathcal{D}_{\text{test}}^0$ and $\mathcal{D}_{\text{test}}^1$ denote the non-member and member subsets of $\mathcal{D}_{\text{test}}$, respectively.

For the adjusted moments estimator experiment in Figure 6, we again split the dataset into two equal halves, assigning one to $\mathcal{D}_{\text{test}}$ and the other to $\mathcal{D}_{\text{cal}}$.

## F ADDITIONAL RESULTS

### F.1 EVALUATION OF SUBTRACTION ESTIMATOR

We evaluated our subtraction estimator using GPT-NeoX-20B on the ArXivTection dataset. The results in Table 4 show a consistently negative bias ($\mathbb{E}[\hat{\pi}_{\text{sub}}] - \pi_{\text{test}} \leqslant 0$) for all detection scores and tested values of $\pi_{\text{test}}$. This confirms that our estimator is conservative, as shown in Equation (11). Since the bias was always negative, we report its magnitude in the table.

### F.2 RESULT ON VISION LANGUAGE MODELS

We further evaluate our method on training data identification for vision–language models (VLMs). Following prior work (Li et al., 2024b), we conduct experiments on the VL-MIA/Flickr and VL-MIA/DALL-E using LLaVA-1.5 (Liu et al., 2023). The detection score $T(X)$ is computed via the MaxRényi-K% statistic (Li et al., 2024b), with hyperparameters $K = 10$ and $\gamma = 0.5$ (see

Table 4: Negative bias and Mean Squared Error (MSE) of the subtraction estimator, evaluated with GPT-NeoX-20B on the ArxivTection dataset.

| Method | $\pi = 0.1$ | | $\pi = 0.3$ | | $\pi = 0.5$ | | $\pi = 0.7$ | | $\pi = 0.9$ | |
|--------|------|-----|------|-----|------|-----|------|-----|------|-----|
| | \|Bias\| | MSE | \|Bias\| | MSE | \|Bias\| | MSE | \|Bias\| | MSE | \|Bias\| | MSE |
| Perplexity | 0.030 | 0.082 | 0.068 | 0.064 | 0.101 | 0.047 | 0.125 | 0.038 | 0.146 | 0.029 |
| Zlib | 0.011 | 0.085 | 0.080 | 0.072 | 0.108 | 0.051 | 0.144 | 0.045 | 0.170 | 0.038 |
| MIN-K% | 0.033 | 0.087 | 0.045 | 0.062 | 0.077 | 0.039 | 0.090 | 0.028 | 0.100 | 0.017 |
| M-Entropy | 0.017 | 0.089 | 0.067 | 0.064 | 0.103 | 0.049 | 0.136 | 0.039 | 0.149 | 0.030 |

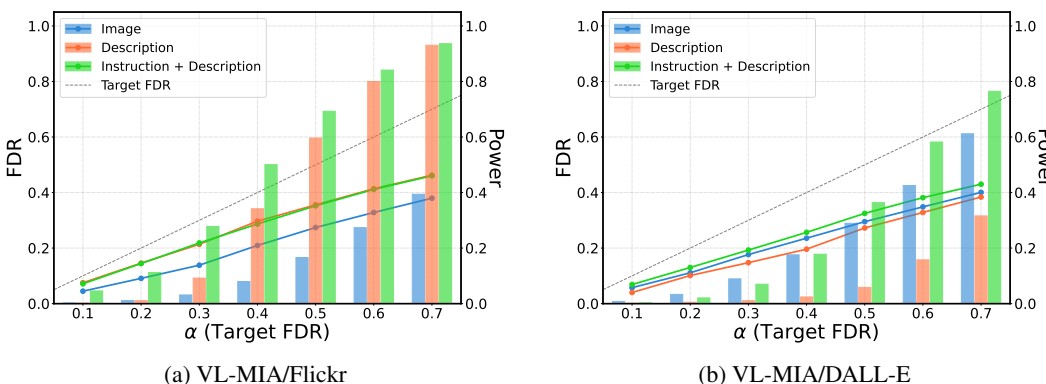

(a) VL-MIA/Flickr    (b) VL-MIA/DALL-E

Figure 7: FDR (solid lines) and power (bars) achieved by our method on LLaVA-1.5. Results are based on the MaxR'enyi-K% score computed from three types of inputs: image embeddings, generated descriptions, and instructions concatenated with descriptions.

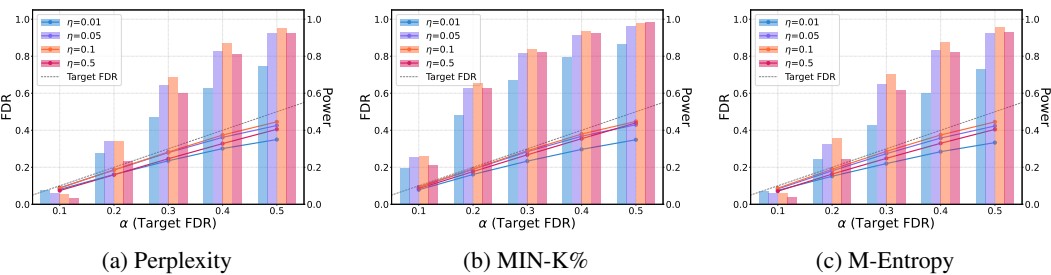

(a) Perplexity    (b) MIN-K%    (c) M-Entropy

Figure 8: FDR (solid lines) and power (bars) achieved by our method varying the hyperparameter $\eta$ that is used to estimate data usage proportion $\pi_{\text{test}}$.

Equation (3)). As shown in Figure 7, our method consistently controls the false discovery rate (FDR), with the realized FDR remaining below the nominal level $\alpha$ across all settings.

### F.3 ABLATION STUDY ON THE P-VALUE SCALING

This appendix section contains a more detailed comparison of our scaling method against the vanilla approach. Table 5 shows the full empirical FDR results on WikiMIA. Furthermore, we present a comprehensive evaluation on the ArXivTection dataset, comparing detection power in Table 7 and empirical FDR in Table 6.

Table 5: FDR on the WikiMIA dataset. We evaluate our method using scaled p-values (***Ours***) against the baseline using original p-values (Vanilla) across various LLMs and detection scores at different target FDR levels ($\alpha$).

| Model | Method | $\alpha = 0.1$ | | $\alpha = 0.2$ | | $\alpha = 0.3$ | | $\alpha = 0.4$ | | $\alpha = 0.5$ | |
|---|---|---|---|---|---|---|---|---|---|---|---|
| | | Vanilla | *Ours* | Vanilla | *Ours* | Vanilla | *Ours* | Vanilla | *Ours* | Vanilla | *Ours* |
| NeoX-20B | Perplexity | 0.05 | 0.07 | 0.09 | 0.15 | 0.15 | 0.22 | 0.19 | 0.29 | 0.24 | 0.35 |
| | Zlib | 0.05 | 0.08 | 0.09 | 0.17 | 0.14 | 0.25 | 0.19 | 0.32 | 0.24 | 0.39 |
| | MIN-K% | 0.05 | 0.08 | 0.10 | 0.17 | 0.15 | 0.25 | 0.20 | 0.32 | 0.25 | 0.38 |
| | M-Entropy | 0.05 | 0.07 | 0.09 | 0.15 | 0.15 | 0.23 | 0.20 | 0.30 | 0.25 | 0.36 |
| LLaMA-7B | Perplexity | 0.04 | 0.08 | 0.09 | 0.17 | 0.14 | 0.26 | 0.18 | 0.33 | 0.24 | 0.39 |
| | Zlib | 0.04 | 0.08 | 0.09 | 0.16 | 0.14 | 0.24 | 0.19 | 0.32 | 0.24 | 0.38 |
| | MIN-K% | 0.05 | 0.07 | 0.09 | 0.15 | 0.15 | 0.22 | 0.20 | 0.29 | 0.25 | 0.35 |
| | M-Entropy | 0.06 | 0.09 | 0.09 | 0.17 | 0.15 | 0.24 | 0.19 | 0.31 | 0.25 | 0.37 |
| Pythia-6.9B | Perplexity | 0.05 | 0.07 | 0.09 | 0.13 | 0.14 | 0.20 | 0.19 | 0.26 | 0.24 | 0.32 |
| | Zlib | 0.05 | 0.06 | 0.09 | 0.13 | 0.14 | 0.20 | 0.19 | 0.27 | 0.24 | 0.33 |
| | MIN-K% | 0.05 | 0.06 | 0.10 | 0.14 | 0.15 | 0.21 | 0.20 | 0.27 | 0.25 | 0.32 |
| | M-Entropy | 0.04 | 0.06 | 0.09 | 0.13 | 0.15 | 0.20 | 0.20 | 0.27 | 0.25 | 0.33 |

Table 6: FDR on the ArXivTection dataset. We evaluate our method using scaled p-values (***Ours***) against the baseline using original p-values (Vanilla) across various LLMs and detection scores at different target FDR levels ($\alpha$).

| Model | Method | $\alpha = 0.1$ | | $\alpha = 0.2$ | | $\alpha = 0.3$ | | $\alpha = 0.4$ | | $\alpha = 0.5$ | |
|---|---|---|---|---|---|---|---|---|---|---|---|
| | | Vanilla | *Ours* | Vanilla | *Ours* | Vanilla | *Ours* | Vanilla | *Ours* | Vanilla | *Ours* |
| NeoX-20B | Perplexity | 0.05 | 0.09 | 0.09 | 0.19 | 0.15 | 0.28 | 0.20 | 0.36 | 0.25 | 0.43 |
| | Zlib | 0.05 | 0.09 | 0.09 | 0.17 | 0.14 | 0.26 | 0.19 | 0.35 | 0.25 | 0.42 |
| | MIN-K% | 0.04 | 0.09 | 0.10 | 0.19 | 0.15 | 0.28 | 0.20 | 0.37 | 0.25 | 0.43 |
| | M-Entropy | 0.05 | 0.08 | 0.09 | 0.18 | 0.15 | 0.27 | 0.20 | 0.36 | 0.26 | 0.42 |
| LLaMA-7B | Perplexity | 0.06 | 0.10 | 0.11 | 0.18 | 0.15 | 0.27 | 0.19 | 0.35 | 0.25 | 0.42 |
| | Zlib | 0.07 | 0.10 | 0.11 | 0.17 | 0.16 | 0.26 | 0.22 | 0.33 | 0.25 | 0.39 |
| | MIN-K% | 0.04 | 0.08 | 0.09 | 0.18 | 0.15 | 0.28 | 0.19 | 0.36 | 0.25 | 0.42 |
| | M-Entropy | 0.05 | 0.09 | 0.10 | 0.17 | 0.14 | 0.26 | 0.18 | 0.35 | 0.24 | 0.42 |
| Pythia-6.9B | Perplexity | 0.05 | 0.08 | 0.08 | 0.17 | 0.14 | 0.27 | 0.20 | 0.35 | 0.25 | 0.42 |
| | Zlib | 0.05 | 0.09 | 0.09 | 0.17 | 0.14 | 0.26 | 0.19 | 0.34 | 0.25 | 0.41 |
| | MIN-K% | 0.04 | 0.10 | 0.10 | 0.20 | 0.15 | 0.30 | 0.20 | 0.38 | 0.25 | 0.44 |
| | M-Entropy | 0.05 | 0.08 | 0.09 | 0.17 | 0.14 | 0.27 | 0.20 | 0.35 | 0.26 | 0.42 |

### F.4 SENSITIVE ANALYSIS TO THE HYPERPARAMETER $\eta$

To investigate the impact on the hyperparameter $\eta$, we conduct experiments with GPT-NeoX-20B on ArxivTection varing $\eta$ in { 0.01, 0.05, 0.1, 0.5 }. The results in Figure 8 show that our method robustly controls the FDR across all tested values of $\eta$.

### F.5 EVALUATION ON REFERENCE-BASED DETECTION SCORE

To demonstrate the versatility of our approach, we further evaluate its integration with reference-based detection scores. Specifically, for pretraining data of LLMs, we experiment on the ArXiv-Tection dataset using GPT-NeoX-20B, employing the Ref-Small score (Carlini et al., 2021) with GPT-Neo-125M as the reference model. As for reference-based MIAs, we train a ResNet-50 (He et al., 2016) on CIFAR-10 (Krizhevsky et al., 2009) and apply the offline LiRA attack (Carlini et al., 2022) using eight reference models. The results in Figure 9 demonstrate that our method consis-

Table 7: Comparison of detection power on the ArXivTection dataset. We evaluate our method using scaled p-values (***Ours***) against the baseline using original p-values (Vanilla) across various LLMs and detection scores at different target FDR levels ($\alpha$). Higher power is highlighted in **bold**.

| Model | Method | $\alpha = 0.1$ Vanilla | ***Ours*** | $\alpha = 0.2$ Vanilla | ***Ours*** | $\alpha = 0.3$ Vanilla | ***Ours*** | $\alpha = 0.4$ Vanilla | ***Ours*** | $\alpha = 0.5$ Vanilla | ***Ours*** |
|---|---|---|---|---|---|---|---|---|---|---|---|
| **NeoX-20B** | Perplexity | 0.01 | **0.06** | 0.06 | **0.34** | 0.18 | **0.65** | 0.40 | **0.83** | 0.65 | **0.93** |
| | Zlib | 0.00 | **0.01** | 0.01 | **0.08** | 0.02 | **0.25** | 0.06 | **0.49** | 0.16 | **0.70** |
| | MIN-K% | 0.05 | **0.25** | 0.28 | **0.63** | 0.55 | **0.82** | 0.70 | **0.91** | 0.80 | **0.96** |
| | M-Entropy | 0.02 | **0.06** | 0.06 | **0.33** | 0.19 | **0.65** | 0.42 | **0.83** | 0.67 | **0.93** |
| **LLaMA-7B** | Perplexity | 0.00 | **0.00** | 0.00 | **0.04** | 0.01 | **0.19** | 0.02 | **0.46** | 0.05 | **0.72** |
| | Zlib | 0.00 | **0.00** | 0.00 | **0.01** | 0.00 | **0.07** | 0.01 | **0.19** | 0.01 | **0.38** |
| | MIN-K% | 0.00 | **0.01** | 0.01 | **0.17** | 0.06 | **0.45** | 0.16 | **0.71** | 0.37 | **0.87** |
| | M-Entropy | 0.00 | **0.00** | 0.00 | **0.04** | 0.01 | **0.19** | 0.02 | **0.48** | 0.06 | **0.73** |
| **Pythia-6.9B** | Perplexity | 0.00 | **0.03** | 0.02 | **0.22** | 0.09 | **0.54** | 0.27 | **0.77** | 0.54 | **0.90** |
| | Zlib | 0.00 | **0.01** | 0.01 | **0.05** | 0.01 | **0.18** | 0.03 | **0.37** | 0.08 | **0.60** |
| | MIN-K% | 0.02 | **0.20** | 0.19 | **0.57** | 0.46 | **0.79** | 0.62 | **0.90** | 0.72 | **0.96** |
| | M-Entropy | 0.01 | **0.03** | 0.03 | **0.23** | 0.11 | **0.55** | 0.30 | **0.78** | 0.56 | **0.90** |

Table 8: Average Hamming Distance on the WikiMIA dataset (test set size $m = 387$). The results are evaluated using our method (subtraction estimator) across various LLMs and detection scores at different target FDR levels ($\alpha$). A lower value indicates better overall set recovery.

| Model | Score | $\alpha = 0.1$ | $\alpha = 0.15$ | $\alpha = 0.2$ | $\alpha = 0.25$ | $\alpha = 0.3$ | $\alpha = 0.35$ | $\alpha = 0.4$ |
|---|---|---|---|---|---|---|---|---|
| **NeoX-20B** | Perplexity | 183 | 175 | 166 | 157 | 150 | 144 | 140 |
| | Zlib | 184 | 174 | 164 | 155 | 147 | 141 | 138 |
| | MIN-K% | 172 | 159 | 148 | 140 | 135 | 132 | 131 |
| | M-Entropy | 182 | 174 | 167 | 159 | 152 | 146 | 142 |
| **LLaMA-7B** | Perplexity | 189 | 184 | 176 | 166 | 158 | 152 | 147 |
| | Zlib | 189 | 186 | 181 | 173 | 164 | 156 | 149 |
| | MIN-K% | 188 | 183 | 177 | 172 | 165 | 159 | 153 |
| | M-Entropy | 191 | 188 | 184 | 178 | 171 | 164 | 158 |
| **Pythia-6.9B** | Perplexity | 190 | 187 | 183 | 178 | 172 | 166 | 160 |
| | Zlib | 190 | 187 | 182 | 176 | 171 | 164 | 159 |
| | MIN-K% | 186 | 179 | 173 | 166 | 160 | 154 | 149 |
| | M-Entropy | 189 | 185 | 181 | 176 | 171 | 166 | 161 |

tently improves power over the vanilla baseline while maintaining valid FDR control, confirming that it is agnostic to the underlying detection scoring mechanism.

## F.6    EVALUATION ON FINE-TUNING SETTING

Beyond the pre-training setting, we further investigate the effectiveness of our scaled p-value procedure in a fine-tuning scenario. Specifically, we fine-tune GPT-NeoX-20B on a subset of the WikiMIA dataset, comprised of 50% of the original non-member data. As demonstrated in 10, our method consistently achieves higher statistical power compared to the vanilla baseline while maintaining strict FDR control.

Table 9: Average Hamming Distance on the ArXivTection dataset (test set size $m = 387$). The results are evaluated using our method (subtraction estimator) across various LLMs and detection scores at different target FDR levels ($\alpha$). A lower value indicates better overall set recovery.

| Model | Score | $\alpha = 0.1$ | $\alpha = 0.15$ | $\alpha = 0.2$ | $\alpha = 0.25$ | $\alpha = 0.3$ | $\alpha = 0.35$ | $\alpha = 0.4$ |
|---|---|---|---|---|---|---|---|---|
| **NeoX-20B** | Perplexity | 365 | 329 | 285 | 252 | 234 | 234 | 248 |
| | Zlib | 384 | 378 | 363 | 344 | 325 | 310 | 301 |
| | MIN-K% | 299 | 235 | 201 | 190 | 196 | 214 | 239 |
| | M-Entropy | 365 | 331 | 289 | 251 | 231 | 231 | 245 |
| **LLaMA-7B** | Perplexity | 386 | 382 | 374 | 359 | 342 | 322 | 305 |
| | Zlib | 387 | 386 | 384 | 380 | 370 | 359 | 349 |
| | MIN-K% | 382 | 365 | 337 | 305 | 279 | 267 | 268 |
| | M-Entropy | 386 | 384 | 375 | 360 | 339 | 318 | 303 |
| **Pythia-6.9B** | Perplexity | 378 | 356 | 320 | 282 | 256 | 246 | 250 |
| | Zlib | 385 | 380 | 370 | 357 | 342 | 326 | 316 |
| | MIN-K% | 319 | 255 | 220 | 207 | 211 | 228 | 252 |
| | M-Entropy | 377 | 354 | 316 | 280 | 252 | 242 | 249 |

Table 10: Comparison of FDR and Power on the WikiMIA dataset. The results are presented in the format KTD / Ours across various target FDR levels ($\alpha$).

| Model | Metric | $\alpha = 0.1$ | $\alpha = 0.15$ | $\alpha = 0.2$ | $\alpha = 0.25$ | $\alpha = 0.3$ |
|---|---|---|---|---|---|---|
| **GPT2** | FDR | 0.195 / 0.098 | 0.276 / 0.148 | 0.337 / 0.197 | 0.362 / 0.247 | 0.362 / 0.295 |
| | Power | 0.957 / 0.797 | 0.984 / 0.908 | 0.994 / 0.953 | 0.995 / 0.976 | 0.995 / 0.988 |
| **Pythia** | FDR | 0.283 / 0.099 | 0.344 / 0.149 | 0.381 / 0.198 | 0.381 / 0.248 | 0.381 / 0.296 |
| | Power | 0.994 / 0.993 | 0.994 / 0.994 | 0.994 / 0.994 | 0.994 / 0.994 | 0.994 / 0.994 |
| **GPT-Neo** | FDR | 0.288 / 0.100 | 0.354 / 0.150 | 0.382 / 0.200 | 0.382 / 0.251 | 0.382 / 0.300 |
| | Power | 1.000 / 0.998 | 1.000 / 0.999 | 1.000 / 0.999 | 1.000 / 1.000 | 1.000 / 1.000 |

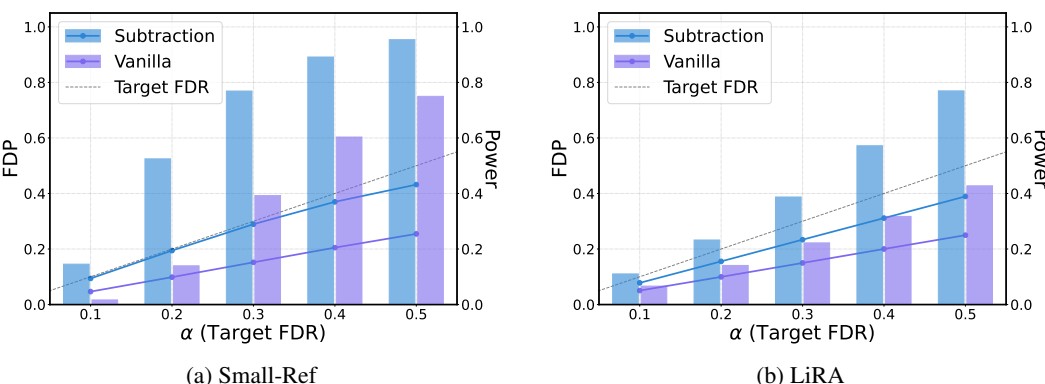

(a) Small-Ref                    (b) LiRA

Figure 9: Performance evaluation using reference-based detection scores. We evaluate (a) the Small-Ref score on the ArXivTection dataset and (b) the LiRA score on CIFAR-10. Each plot shows the realized FDR (solid lines) and power (bars) for a detection score.

## F.7 ANALYSIS OF SET-RECOVERY METRICS

To provide a comprehensive evaluation, we report the average Hamming distance of experiment in Figure 1 between the predicted membership set and the ground truth as set recovery performance. Specifically, for a test set of size $m$, the Hamming distance is defined as the total count of classi-

Table 11: Comparison of FDR and Power on the XSum dataset. The results are presented in the format KTD / Ours across various target FDR levels ($\alpha$).

| Model | Metric | $\alpha = 0.1$ | $\alpha = 0.15$ | $\alpha = 0.2$ | $\alpha = 0.25$ | $\alpha = 0.3$ |
|-------|--------|------|------|------|------|------|
| GPT2 | FDR | 0.169 / 0.099 | 0.283 / 0.149 | 0.344 / 0.199 | 0.388 / 0.249 | 0.388 / 0.299 |
|  | Power | 0.983 / 0.960 | 0.995 / 0.979 | 0.997 / 0.988 | 0.999 / 0.993 | 0.999 / 0.996 |
| Pythia | FDR | 0.104 / 0.100 | 0.161 / 0.150 | 0.219 / 0.200 | 0.279 / 0.250 | 0.330 / 0.300 |
|  | Power | 1.000 / 1.000 | 1.000 / 1.000 | 1.000 / 1.000 | 1.000 / 1.000 | 1.000 / 1.000 |
| GPT-Neo | FDR | 0.116 / 0.100 | 0.214 / 0.150 | 0.284 / 0.200 | 0.342 / 0.250 | 0.371 / 0.300 |
|  | Power | 0.999 / 0.999 | 1.000 / 0.999 | 1.000 / 1.000 | 1.000 / 1.000 | 1.000 / 1.000 |

Table 12: Comparison of FDR and Power on the BBC Real Time dataset. The results are presented in the format KTD / Ours across various target FDR levels ($\alpha$).

| Model | Metric | $\alpha = 0.1$ | $\alpha = 0.15$ | $\alpha = 0.2$ | $\alpha = 0.25$ | $\alpha = 0.3$ |
|-------|--------|------|------|------|------|------|
| GPT2 | FDR | 0.087 / 0.100 | 0.118 / 0.149 | 0.156 / 0.199 | 0.199 / 0.249 | 0.235 / 0.299 |
|  | Power | 0.925 / 0.950 | 0.969 / 0.980 | 0.982 / 0.991 | 0.991 / 0.994 | 0.993 / 0.996 |
| Pythia | FDR | 0.071 / 0.100 | 0.091 / 0.150 | 0.113 / 0.200 | 0.137 / 0.250 | 0.185 / 0.300 |
|  | Power | 1.000 / 1.000 | 1.000 / 1.000 | 1.000 / 1.000 | 1.000 / 1.000 | 1.000 / 1.000 |
| GPT-Neo | FDR | 0.085 / 0.100 | 0.118 / 0.150 | 0.155 / 0.200 | 0.201 / 0.250 | 0.259 / 0.300 |
|  | Power | 0.999 / 1.000 | 1.000 / 1.000 | 1.000 / 1.000 | 1.000 / 1.000 | 1.000 / 1.000 |

fication errors: Hamming $= \mathbb{E}\left[\sum_{j=1}^{m}\left(\mathbb{1}\{j \in \mathcal{S}, M_{n+j} = 0\} + \mathbb{1}\{j \notin \mathcal{S}, M_{n+j} = 1\}\right)\right]$. Table 8 and Table 9 present the average Hamming distance for WikiMIA and ArXivTection, respectively. We observe that the Hamming distance is higher at lower target FDR levels (e.g., $\alpha = 0.1$). This is primarily driven by the false negative count, as our method prioritizes strict FDR control, resulting in conservative power when the target FDR level is low.

### F.8 ADDITIONAL RESULTS ON KTD COMPARISON

In this section, we provide supplementary details for the comparison experiment with the Knockoff inference-based Training data Detector (KTD), supporting Figure 2 and Figure 3. The FDR and power results across various target FDR levels are reported in Table 10, Table 11, and Table 12 for the WikiMIA, XSum, and BBC Real Time datasets, respectively. The results demonstrate that our method consistently controls FDR. Moreover, in the settings where KTD also achieves valid FDR control, our method yields higher power.

## G FROM SINGLE-POINT INFERENCE TO MULTIPLE TESTING: THE MIA DILEMMA

### G.1 THE IDEAL: A RIGOROUS SINGLE-POINT TEST

Given a data point $X$ and a trained target model $\theta_1$, membership inference attacks (MIAs) (Shokri et al., 2017; Yeom et al., 2018; Salem et al., 2019) aim to identify whether $X$ is one of the members in the training set $\mathcal{D}_{\text{train}}$. This type of privacy attack is often modeled as a statistical hypothesis testing problem (Ye et al., 2022; Carlini et al., 2022; Bertran et al., 2024; Zarifzadeh et al., 2024):

$$H_0 : X \notin \mathcal{D}_{\text{train}} \quad \text{v.s.} \quad H_1 : X \in \mathcal{D}_{\text{train}}. \tag{20}$$

Here, the null hypothesis ($H_0$) posits that $X$ is a non-member, meaning it was drawn from the same underlying data distribution as $\mathcal{D}_{\text{train}}$ but was not included in it. Conversely, the alternative hypothesis ($H_1$) posits that $X$ is a member of the training set.

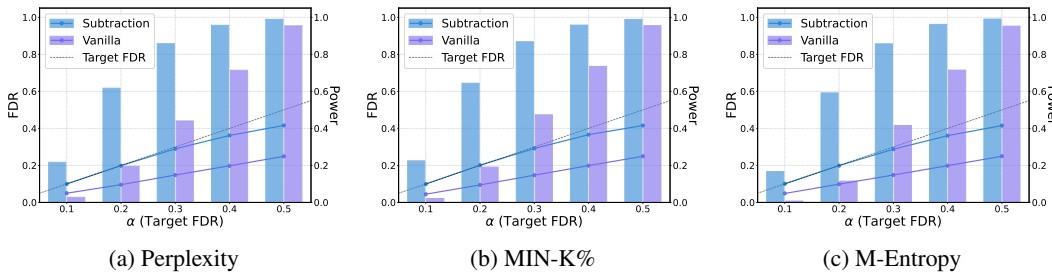

(a) Perplexity        (b) MIN-K%        (c) M-Entropy

Figure 10: Performance evaluation in the fine-tuning setting (WikiMIA / GPT-NeoX-20B). We compare our proposed method against the vanilla baseline using three detection scores. Each plot shows the realized FDR (solid lines) and power (bars) for a detection score.

To reject the null hypothesis, we compute membership inference attack (MIA) scores, such as the model's loss or confidence on data point $X$. For instance, let $T(X, \theta)$ represent the loss of $X$ produced by model $\theta$, then we can reject $H_0$ when $T(X, \theta) \leqslant \tau$ [1]. To control the type I error, which is identical to the false positive rate (FPR) at the sample level, we choose $\tau$ such that

$$\Pr_{\theta \sim \Theta_0}[T(X, \theta) \leqslant \tau] \leqslant \alpha \tag{21}$$

where $\Theta_0$ is the distribution over model parameters when trained on datasets that do not contain $X$ (under $H_0$), $\tau$ is the threshold used to define the rejection region. In practice, sampling $\theta$ typically requires training multiple reference models (Ye et al., 2022) or constructing Bayesian neural networks from a single reference model (Liu et al., 2025).

However, estimating this reject region is challenging for large-scale models, such as ChatGPT or DALL-E, due to limited access to the training data distribution and the training algorithm, compounded by the prohibitively high computational cost of training (Zhang et al., 2025b).

### G.2 THE REALITY: A HEURISTIC WITH A CONCEPTUAL FLAW

Several studies on MIAs applied to large language models and vision-language models (Fu et al., 2024; Carlini et al., 2021; Shi et al., 2024; Zhang et al., 2025a) report true positive rates (TPR) at low FPRs using only MIA scores from the target model. This is typically achieved via a heuristic method: a single score threshold $\tau$ is determined on a calibration set of non-members to control the average FPR(Ye et al., 2022). Formally, this metric can be written as the expected Type I error:

$$\mathrm{FPR} = \mathbb{E}_X\left[\Pr[T(X, \theta_1) \leqslant \tau \mid H_0]\right]. \tag{22}$$

Herein lies a subtle but critical conceptual flaw. The average FPR conflates the overall error rate with the per-hypothesis Type I error rate, $\Pr[T(X, \theta_1) \leqslant \tau \mid H_0]$. As an average metric, it does not provide a probabilistic guarantee for any single inference. A low average FPR can mask a much higher error rate for specific subgroups of data, offering no reliable evidence for any individual decision.

Furthermore, in practical scenarios like copyright infringement litigation, where the goal is to identify a reliable set of members from many candidates, the average FPR remains inappropriate. This scenario is a classic multiple testing problem. In this setting, the objective is not to manage an average error rate over all non-members, but to control the fraction of incorrect claims among the discoveries made. This is precisely the quantity measured by the False Discovery Rate (FDR), the statistically sound tool for this task.

The distinction between these metrics is critical in practice. Consider an audit of one million candidates containing only 1,000 true members. A method with seemingly excellent performance, such as a 0.1% average FPR and 80% TPR, would nevertheless yield approximately 999 false positives alongside 800 true positives. The resulting set's FDR would be an untenable 55.5%, meaning over

---

[1]Lower loss suggests that $X$ was likely to be a part of the training set.

half the presented evidence is incorrect. This starkly demonstrates that average FPR is a fundamentally inadequate and misleading metric for ensuring the credibility of membership inference claims in a legal or auditing context.

