# OpenReview forum: "High-Power Training Data Identification with Provable Statistical Guarantees"
_ICLR.cc/2026/Conference — Submitted to ICLR 2026_

### Official Review · Reviewer_jywB · 2025-10-26

**Soundness:** 3
**Presentation:** 2
**Contribution:** 2
**Rating:** 6
**Confidence:** 3

**Summary:**

This paper introduces Provable Training Data Identification (PTDI), a statistical framework for identifying whether a sample was part of a model’s training data while controlling the false discovery rate (FDR) under finite-sample guarantees. The method leverages conformal prediction to compute $p$-values using a calibration set of known non-members and applies the Benjamini–Hochberg procedure to select positives with provable statistical control. PTDI is evaluated on LLMs and VLMs, showing stronger identification power than heuristic membership inference baselines.

**Strengths:**

- The paper presents a clear and principled statistical framework that brings formal guarantees to training data identification, addressing a key gap in existing heuristic or ad hoc auditing methods.

- The integration of conformal $p$-value calibration with FDR control is elegant and theoretically sound, and the approach demonstrates consistent empirical performance across both LLMs and VLMs.

**Weaknesses:**

- The method’s theoretical guarantees, including valid conformal p-values and provable FDR control, hinge on the calibration non-members being exchangeable with test non-members. In practice, this assumption is unverifiable and unlikely to hold in realistic auditing scenarios with heterogeneous or evolving data distributions. The authors acknowledge this limitation, but it substantially weakens the “provable” claim in real-world applications.

- The paper’s most powerful variant, the adjusted moment estimator, requires access to a calibration set of confirmed members and non-members. This assumption is circular in the intended auditing context, where the goal is to discover unknown members. This makes the paper's best-performing method practically infeasible for its primary use case.

**Questions:**

N/A

---

> ### Author Response · Authors · 2025-11-24
> **Response to Reviewer jywB**
>
> Thank you for your valuable feedback and positive assessment. We appreciate the opportunity to clarify your two main concerns.
>
>
> ### **1. On the Data Assumption [W1]**
> Yes, our theoretical guarantees rely on the assumption that the calibration and test non-members share the same distribution, which is practical in certain auditing scenarios. For instance, in copyright litigation, a painter can use their unreleased works (confirmed non-members) as the calibration set to test whether their published paintings were used in training. We'd like to emphasize that this work presents a novel method to provide provable theoretical guarantees for identifying the membership in benchmarks, with a relatively weak assumption. While heterogeneous test data can be a challenge for our method (as discussed in our limitations), we believe our method can inspire more specific-design methods to handle this setting in the future.
>
> ### **2. Clarification on the Adjusted Moment Estimator [W2]**
>
> Thank you for raising this concern. We would like to clarify that the adjusted moment estimator $\pi_{\text{mom}}$ is proposed for the scenario where a calibration set containing a mix of confirmed members and non-members is available. In particular, the test set and calibration set are **totally disjoint**, so this assumption is not "circular".
>
> Moreover, this assumption is practical in certain scenarios. A wide range of works [1, 2, 3] establish that some public corpora (e.g., Wikipedia, The Pile) are widely used for pre-training LLMs. For example, the benchmark WikiMIA [4] collected articles created before 2017 as member data. In such cases, samples from these known public sources can serve as the "confirmed members" to calibrate the estimator, thereby enabling application of our adjusted moment estimator.
>
>
> [1] "Investigating Data Contamination in Modern Benchmarks
> for Large Language Models" (ACL, 2024)
>
> [2] "Rethinking Benchmark and Contamination for Language Models with Rephrased Samples" (ArXiv, 2023)
>
> [3] "Do Membership Inference Attacks Work on Large Language Models?" (COLM, 2024)
>
> [4] "Detecting Pretraining Data from Large Language Models" (Neurips, 2024)

---

### Official Review · Reviewer_GF9Q · 2025-10-31

**Soundness:** 3
**Presentation:** 3
**Contribution:** 2
**Rating:** 4
**Confidence:** 4

**Summary:**

The paper proposes a method for training-data identification that constructs p-values via conformal inference using a post-cutoff non-member population set. The procedure can be plugged into existing MIA scores, aiming to provide finer-grained data-provenance information than a binary decision. The method relies on the assumption that data points are i.i.d. samples from a unified population distribution, meaning each point has the same probability of being selected into training and the same MIA error characteristics (i.e., identical TPR/FPR across samples). With a non-member calibration set (under the same i.i.d. assumptions), the method can control the FDR, which the paper defines as the expectation of 1-precision. Experiments on vision-language and language domains show its applicability.

**Strengths:**

1. Problem motivation is meaningful. The paper studies data provenance/auditing questions in the context of large models, which is relevant for practical concerns such as copyright verification.
2. The experiments cover both language and vision–language models, and evaluate across multiple widely-used (MIA) signals (e.g., perplexity, MIN-K%, Rényi entropy, knockoff-based score), showing its realistic applicability.
3. The paper is clearly written. The method is straightforward to understand and principled.

**Weaknesses:**

I will likely increase my score if the necessary experiments and explanations are added.
1. **Key assumptions are less realistic, especially in the data provenance problem discussed here.** The core idea of this method is to apply scaled p-values in conformal inference, with the scaling term estimated via data usage proportion. However, Eq.10 in Sec.3.2 relies on two strong assumptions: (i) identical detection behavior (e.g., MIA errors) across samples, i.e., $P(t|M=1)=P(t|M_i=1)$ for all data $i$, and (ii) i.i.d. data sampling $P(M_i=1)=\pi_{\text{test}}$. In practice, high-value/copyrighted data will not be sampled uniformly, and detection scores (e.g., MIA signals) are known to be heterogeneous across samples (different memorization/detectability). Therefore, (1) these assumptions should be clearly stated and justified, and (2) experiments with intentionally biased sampling should be included to evaluate robustness.
2. **Extremely low power when the target FDR is small.** In Figure 1, when the target FDR is low, empirical FDR stays below the target, but power becomes extremely low. This corresponds to cases like: there are 100 test samples with 99 true training points, yet the method only identifies *one* of them (so FDR=0) and misses the rest (so Power = 1/99). Is this an intended behavior? It would be more informative to report a true set-recovery metric (e.g., Hamming distance between the ground-truth membership vector and the predicted one).

**Questions:**

- What is the connection to DUCI [1]?

Sec. 3.2 seems closely related to a prior work DUCI, which also integrates with arbitrary detection scores. For example, Eq. 10 appears adapted from Eq. 5 in DUCI, the following integral here matches the expectation there, and the core result used in Theorem 1 $\mathbb{E}[\frac{1-\pi_{test}}{1-\hat \pi_{sub}}] \leq 1$ is the unbiasedness $\mathbb{E}[\hat \pi_{sub} - \pi_{test}]\le 0$ in DUCI. If I understand correctly, the main difference is that Eq. 10 assumes homogeneous sampling probability/MIA bias, which allows integration directly over the data distribution (removing the reference-model dimension)? It would be helpful if the authors could explain the differences, but extending the estimator from statistical proportion estimation to set prediction is already a useful contribution.

- Minor: Line 131: Missing period after “candidate training samples”

[1] How much of my dataset did you use? Quantitative Data Usage Inference in Machine Learning. ICLR’25

---

> ### Author Response · Authors · 2025-11-24
> **Response to Reviewer GF9Q**
>
> Thank you for your insightful feedback on our manuscript. We appreciate the constructive comments and provide detailed responses to your concerns below.
>
> ### **1. Clarification on the Data Assumptions [W1]**
> Thank you for pointing out the potential misunderstanding regarding the assumptions for our method. We clarify that our method does **NOT** rely on the two assumptions identified by the reviewer. We respond to each assumption below.
>
>
> - **The “identical detection behavior” assumption**: We would like to clarify that $p(t|M=1)$ is defined as the marginal detection score distribution over the entire training data population. It can be viewed as marginalizing over all possible data points X
> $$p(t|M=1) = \int p(t|M=1, X)p(X|M=1) dX$$
> As the reviewer noted, data can be diverse (e.g., "valuable" vs. "not valuable"). We can illustrate this using the Huber contamination model [1], where the population distribution is expressed as a mixture:
> $$p(t|M=1) =   p(t|\text{valuable}, M=1) p(\text{valuable}|M=1)+  p(t|\text{not valuable}, M=1) p(\text{not valuable}|M=1)$$
>
>     This formulation does not assume all training data have identical detection behavior.
>
>
> - **The "uniform data sampling" assumption**: For a given test set  $\mathcal{D}\_{\text{test}}$, $\pi\_{\text{test}}$  is simply defined as the true, fixed proportion of training data within the test set. $P(M_i=1) = \pi\_{\text{test}}$ refers to the probability of an auditor drawing a member sample when randomly selecting from $\mathcal{D}\_{\text{test}}$. Our method is designed to work for any given test set, and $\pi\_{\text{test}}$ is the property of that set that we are estimating. We make no assumptions about how the model's trainer originally sampled their data.
>
>
>
>
> ### **2. Low Power at Low Target FDR [W2]**
>
> Thank you for highlighting this observation. We would like to clarify that this is an **intended consequence of rigorous FDR control** rather than a flaw. With a fixed heuristic score, our method will select fewer examples that are more likely to be members if we expect a lower FDR in the detection (i.e., non-member ratios in the selected subset). This is particularly important in high-stakes scenarios, such as copyright infringement litigation, where a false accusation can be extremely costly both reputationally and financially. Therefore, deliberately maintaining a stringent FDR is not a compromise on recall, but a principled design choice: we prioritize not wrongly accusing even a single innocent (non-infringing) work over maximizing the number of correctly identified infringements. Besides, we added the Hamming distance evaluation in Tables 8 and 9 (Appendix F.7) of the revision as suggested.
>
>
>
> ### **3. Connection to DUCI [Q1]**
> Yes, estimating the member proportion $\pi_{\text{test}}$ in our method is related to DUCI, which we discuss in our Related Work section. We first reply to the questions proposed by the reviewers, respectively.
>
> > Eq. 10 appears adapted from Eq. 5 in DUCI, the following integral here matches the expectation there, and the core result used in Theorem 1 $|\mathbb{E}[\frac{1-\pi\_{test}}{1-\pi\_{sub}}] - 1| \le 0$ is the unbiasedness $\mathbb{E}[\hat{\pi}\_{sub} - \pi\_{test}] \leq 0$ in DUCI.
>
> No, Eq. 10 is not adapted from Eq. 5 in DUCI. Notably, DUCI's Eq. 5 relates TPR and FPR to the member proportion, while our Eq. 10 connects the entire score distribution of test and calibration data to the proportion. Therefore, our method does not require the estimation of TPR or FPR, as well as reference models. For the estimator bias, DUCI focuses on an unbiased estimator of $\pi_{\text{test}}$ while our estimator $\hat{\pi}\_{\text{sub}}$ is intentionally designed to be conservative (biased), tending to underestimate $\pi_{\text{test}}$. This conservative bias is essential for our theoretical guarantee, as shown in the proof of Theorem 1.
>
> > The main difference is that Eq. 10 assumes homogeneous sampling probability/MIA bias, which allows integration directly over the data distribution (removing the reference-model dimension)?
>
>
> **Differences between DUCI and our estimator**: (1) DUCI requires reference models to estimate TPR/FPR while our method does not. This makes our method applicable in the settings of large-scale models, like LLM; (2) Our estimator is intentionally designed to be conservative while DUCI focuses on unbiased estimation; (3) Our estimator is employed to control the provable FDR, whereas DUCI aims to estimate the data usage proportion.
>
>
> ### **4. Typo [Q2]**
>
> Thank you for pointing out the typo. We have corrected the typo in the revised version.
>
>
> [1] "Robust Estimation of a Location Parameter" (Breakthroughs in Statistics, 1992)

---

### Official Review · Reviewer_NNQR · 2025-11-01

**Soundness:** 2
**Presentation:** 2
**Contribution:** 3
**Rating:** 4
**Confidence:** 4

**Summary:**

The paper proposes a scaled p-value transformation that given a calibration set with a small number of ground-truth non-member data, provably transforms any existing MIA score to another MIA that is calibrated (in terms of its FPR on a test set). Experiments on a wide range of pretrained and fine-tuned language model and visual language models show that the transformation consistently makes the derived MIA score calibrated, while also significantly improves the true positive rate of the attack on test set.

**Strengths:**

The derivations of the scaled p-value is novel to the best of my knowledge. Experiments confirm the strong calibration, and even show improving MIA performance across benchmarks, showcasing the effectiveness of the proposed method.

**Weaknesses:**

- The reason for the improved MIA power when applying the proposed scaled p-value transformation not discussed. Section 3.2 only discusses why the derived estimator is calibrated, but did not discuss why it may improve the power of the attack as well. I personally find it very surprising as, if $\hat{\pi}_{test}$ is a test-point independent quantity, then scaled p-value is just a monotonic transformation of the p-values and should not improve attack performance. So the key to improvement appears to be the dependence of the subtraction estimator (12) on the test p-values. I hope the authors could provide more explanations to help understand the reason for the improvement, which to my understanding is the key contribution.
- The authors didn't evaluate any reference model-based MIA baselines. Even in the pertaining setting, it is common to use an off-the-shelf smaller model as reference model ((Carlini et al., 2021), and it would be more convincing to understand if the author's method also improves performance for those stronger MIAs.
- Improved power is only shown for the pretraining setting (Table 1), where the benchmark has temporal shift that may overestimate MIA performance. It would be more convincing if the authors could discuss whether MIA power also improves under scaled p-values for fine-tuning setting.

**Questions:**

See weaknesses.

---

> ### Author Response · Authors · 2025-11-24
> **Response to Reviewer NNQR**
>
> Thank you for your constructive feedback and encouraging comments. Please find our response below.
>
> ### **1. Why our method can improve the Power? [W1]**
> Thank you for this insightful question. We would like to clarify a key distinction: **our method improves the detection power (i.e., the expected proportion of all true members that are correctly identified in $\mathcal{S}$, as defined in Eq. 5) by optimizing the decision threshold while maintaining FDR control**, instead of changing the underlying scoring ranking (i.e., the ROC curve remains unchanged). Notably, the compared method "Vanilla" using unscaled p-values is provably valid but often **overly conservative** (see the empirical FDR in Table 5). This conservatism comes from the standard BH procedure's theoretical bound, which scales with the proportion of true null hypotheses (see Lemma 2). By scaling the p-value, we provide a more precise (and less conservative) adjustment for the BH procedure. This scaling allows our method to "use up" more allotted FDR budget, leading to more discoveries while maintaining FDR control. To rigorously support this, we have added **Theorem 3 in Appendix C.4**, which proves that our scaling procedure strictly improves power compared to the baseline.
>
> ### **2. Evaluation on Reference Model-Based Baselines [W2]**
>
> Thank you for your suggestion. In this work, we focus on the setting where **training multiple reference models is prohibitively expensive or impossible**, such as large language models. However, we agree that demonstrating our method's versatility with reference-model-based scores can be a valuable addition. Thus, we conducted additional experiments to demonstrate that our method can be integrated with any detection score.
>
> For the LLM pre-training setting, we conducted experiments on WikiMIA with GPT-NeoX-20B, using a smaller reference model (GPT-Neo-125M) to calculate the Ref-Small score proposed by Carlini et al. (2021) [1]. The results for power (%) under various target FDR (%) are as follows:
>
>
> | Target FDR      | 10     | 15     | 20     | 25     | 30     |
> |---------------------|--------|--------|--------|--------|--------|
> | Vanilla             | 1.66   | 5.87   | 13.95  | 26.04  | 39.24  |
> | **Ours** | **14.54**  | **33.97**  | **52.47**  | **66.76**  | **76.89**  |
>
>
>
> Regarding the reference-based MIA method, we conduct experiments on CIFAR-10 with model resnet-50, using 8 reference models to calculate LiRA [2] score. The results for power (%) under various target FDR (%) are as follows:
>
>
>
> | Target FDR  (%)        | 10     | 15     | 20     | 25     | 30     |
> |---------------------|--------|--------|--------|--------|--------|
> | Vanilla             | 6.64   | 10.61  | 14.09  | 17.82  | 22.21  |
> | **Ours** | **11.03**  | **16.52**  | **23.25**  | **30.72**  | **38.74**  |
>
>
>
> These results demonstrate that our method can consistently improve power over the vanilla baseline. We present more results in Appendix F.5.
>
>
> ### **3. Evaluation on Fine-tuning Setting [W3]**
>
> Thank you for pointing this out.We conducted experiments using a fine-tuning setup. We fine-tuned GPT-NeoX-20B on a subset of the WikiMIA dataset using the detection score MIN-K% (specifically using samples originally held out as non-members). The table below reports the Power (%) across various target FDR (%) levels, presents our scaling procedure achieves higher power than vanilla. We present more results in Appendix F.6.
>
>
> | Target FDR (%)         | 10  | 15 | 20 | 25 | 30   |
> |---------------------|--------|--------|--------|--------|---------|
> | Vanilla             | 2.13   | 7.95   | 19.09  | 33.47  | 47.33   |
> | **Ours** | **22.47**  | **44.87**  | **64.40**  | **77.86**  | **86.80**   |
>
>
> [1] "Extracting training data from large language models" (USENIX Security, 2021)
>
> [2] "Membership Inference Attacks From First Principles" (S&P, 2022)

---

### Official Review · Reviewer_cobt · 2025-11-01

**Soundness:** 3
**Presentation:** 3
**Contribution:** 2
**Rating:** 2
**Confidence:** 4

**Summary:**

This work is concerned with identifying training data. As has become a recent popular trend, valid conformal p-values are computed using score functions, but before BH is applied, the p-values are scaled via an estimation of the proportion of non-training data in the test set (equiv., the proportion of "nulls", in the language of multiple testing).

A theorem is presented that establishes FDR control, and experiments are conducted to verify this as well as to support the claim that higher power is achieved when the scaling is conducted as oppose to when no scaling is conducted.

**Strengths:**

- Many experiments were conducted to compare the performance of PTDI under various scoring methods, $\alpha$ parameter settings, and on varying datasets.

**Weaknesses:**

- The PTDI algorithm essentially combines conformal p-values with a scaling that is equivalently Storey's modification of the BH algorithm (see "A direct approach to false discovery rates," Storey 2002). As Storey's modification (equivalently, the scaling) was proposed to enhance the power of BH without disrupting its FDR control guarantee, it seems like a clear and natural suggestion to apply this idea to the problem of identifying training data. In other words, the central thesis to this paper is far from novel, and the same applies to their theoretical statistical guarantee for FDR in Theorem 1. And just as in Storey 2002, there isn't a concrete theoretical statement rigorously establishing that higher power is achieved with the Storey modification.

- The comparison in power between PTDI versus KTD is not experimented on as extensively as FDR control. There is but one experiment, the results of which are posted in Figure 3, and I'm afraid they do not make a strong case that PTDI is certifiably a higher power method.

- The Related Works section is a bit sparse. For instance, I think the authors should definitely consider citing "A direct approach to false discovery rates," Storey 2002, considering their scaling of their conformal p-values appears to be precisely mirroring Storey's modification of BH. As well, the authors are leaving out some older sources that could be credited with the source of conformal p-values, like Vovk, V., Gammerman, A., & Shafer, G. (1999) and Vovk, V., Nouretdinov, I., & Gammerman, A. (2003).

**Questions:**

- The PTDI algorithm appears roughly to compute conformal p-values and then apply Storey's modification of the BH algorithm (see "A direct approach to false discovery rates," Storey 2002). Would the authors confirm/clarify the relationship between PTDI and Storey's?

- The paragraph starting on line 041 mentions a handful of recent studies for identifying training data. Can the authors discuss how this paragraph adequately positions your work within the field?

- The comparison in power between PTDI versus KTD is not experimented on as extensively as FDR control. There is but one experiment, the results of which are posted in Figure 3, and I'm afraid they do not make a strong case that PTDI is certifiably a higher power method. Could the authors provide any (more) justification for the claim that PTDI is a higher power method?

- $D_{cal}$ and $D_{test}$ composition: Section E.1 Data Split Setup inadequately details the composition and procedure for curating $D_{cal}$ and $D_{test}$for all datasets employed (WikiMIA, ArxivTection, XSum, BBC Real Time, VL-MIA/Flickr and VL-MIA/DALL-E). Can the authors please provide greater details in this regard? What are the sizes of all of the datasets? Prior to splitting a dataset in half, what percentage of the dataset consists of members/non-members? What are the expected sizes of $D_{cal}$ and $D_{test}$ after splitting a dataset in half?

- A claim is made that PTDI can be readily combined with existing detection methods in both black-box and white-box settings only requiring unseen data as a calibration set. Please confirm whether or not PTDI also needs access to the model outputs on both the calibration and test set as well? Clearly, PTDI was designed for and operates under a "black-box" regime. Consequently, It seems trivial to say that a "black-box" method also operates under a "white-box" regime. Please confirm whether or not PTDI would operate identically under a "white-box" regime? Algorithm 1 indicates that access to the model is required. If the white-box/black-box terminology is to be employed, then I would recommend clarifying that only the outputs of the model on $D_{cal}$ and $D_{test}$ are required, not the model itself. I might consider abandoning the use of the terminology white-box/black-box, in favor of simply stating the requirements of PTDI.

- Can the authors comment on why FDR "levels-off" at a Target FDR of 0.5 in Figure 2? Why does KTD "level-off" sooner than PTDI?  In the event that $D_{test}$ is composed of 50% non-members and 50% members, then a naive detector that simply classifies every test example as a member would achieve an FDR of 0.5.

- Line 367 The section title "How does the scaling procedure affect?" needs and object. An alternative phrasing could be "Effect of the scaling procedure."

- Lines 066-067 percentages are used for FDR. In all other cases percentages are not used for FDR. Recommend picking one format for consistency and clarity.

- Line 1119 subscript "test" on Dtest

---

> ### Author Response · Authors · 2025-11-24
> **Response to Reviewer cobt (1/2)**
>
> Thank you for your constructive and detailed feedback. We provide detailed responses to your concerns below.
>
> ### **1. On the Novelty of PTDI and its Connection to Storey (2002) [W1, Q1]**
>
> Thank you for highlighting the important connection to Storey's (2002) work. We agree that the scaling p-values by the estimated training data proportion $\pi_0$ is similar to Storey's modification of the BH procedure. But we'd like to clarify that the main contribution of this work is proposing an effective method with theoretical guarantees to address the critical problem of training data identification in large-scale models. The contributions of this work are threefold:
>
>
> - **Problem formulation**: We formulate the problem of training data identification, which transfers the classifying single data points (which is a traditional membership inference attack) into a multiple-hypothesis testing framework. This formulation provides a feasible way to construct a provable theoretical guarantee for identifying training data of large models. Before this work, the MIA community claimed it was impossible to prove that a model was trained on specific data [1].
>
> - **Practical method**: Our method is the first to identify training data with a valid theoretical guarantee. Our pipeline is extensive by including the derivation of conformal p-values from existing detection scores and the estimation of the member proportion under distinct scenarios (e.g., $\hat{\pi}\_{sub}$ for the cases where the auditor only has access to the unseen data and $\hat{\pi}\_{mom}$ for scenario where some confirmed members are available). Crucially, our method requires no reference models, no gradients, and no knowledge of the training algorithm.
>
> - **Theoretical guarantee**: We provide rigorous theoretical proofs establishing that our method strictly controls the FDR, thereby providing credible evidence regarding the identified training data.
>
>
> **Why Storey's estimator cannot be directly applied**: We emphasize that Storey's estimator cannot be directly applied to the training data identification scenario because its theoretical framework generally relies on the assumption of **independent p-values** derived from their corresponding hypotheses. In the context of large-scale models, obtaining such independent p-values is intractable, as it typically requires training multiple reference models to estimate the null distribution, which is computationally prohibitive and often impossible due to restricted access to the training algorithm.
>
> Moreover, our conformal p-values are not independent since they are constructed using a shared calibration set. This dependence violates the independence assumption underlying Storey’s estimator, which implies its theoretical guarantee does not hold. In this paper, we construct estimators from detection scores and derive the corresponding properties on propositions (e.g., $\mathbb{E}\left[\frac{1-\pi_{\text{test}}}{1-\hat{\pi}_{\text{sub}}}\right] \leqslant 1$). We leverage this specific property to establish the formal proof for Theorem 1, ensuring that our method achieves valid FDR control specifically under the training data identification task.
>
>
> We have added this discussion to the Related Work section in the updated version.
>
>
> ### **2. Theoretical Guarantee for Improving Power [W1]**
>
> Thank you for this suggestion. To address the concern regarding theoretical guarantees for improving power, we have included Theorem 3 in Appendix C.4. It rigorously proves that scaling p-values with our estimator strictly increases the rejection set size compared to the unscaled baseline, thereby leading to higher power.
>
>
> ### **3. Comparison with KTD [W2, Q3]**
> Thank you for the suggestion on the power comparison with KTD. First, we clarify that **a power comparison is only meaningful when both methods control the FDR successfully.** For example, the power can be even 100% if the realized FDR is equivalent to the non-member proportion of the test set. As shown in Figure 2 of the manuscript, KTD fails to control the FDR on the WikiMIA and XSum datasets. Thus, we only provide the power comparison on BBC Real Time, where KTD successfully controls the FDR ($\alpha \geqslant 0.1$). The results in Figure 3 show that PTDI demonstrates superior power on GPT-2 in the fair comparison. In summary, PTDI is the only method that both **strictly controls FDR** across all settings and demonstrates high power. For reference, detailed experimental results comparing our method with KTD are provided in Tables 10–12 in Appendix F.8, containing the WikiMIA, XSum, and BBC Real Time datasets.

---

> ### Author Response · Authors · 2025-11-24
> **Response to Reviewer cobt (2/2)**
>
> ### **4. Missing Citations in Related Work [W3]**
> Thank you for these valuable suggestions. We agree that Storey (2002) is highly relevant to our p-value scaling procedure and that Vovk et al. (1999, 2003) are the foundation to the development of conformal p-values. We have updated our Related Work and Methodology sections to include and discuss these important citations.
>
> ###  **5. Positioning of Our Work [Q2]**
> Thank you for this question. The paragraph starting at line 074 highlights two key gaps in existing literature:
>
> - **Lack of Theoretical Guarantees**: Prior works have introduced effective detection scores, such as MIN-K% for LLMs (Shi et al., 2024 [2]), MaxRényi-K% for VLMs (Li et al., 2024 [3]), and the fine-tuning deviation metric FSD (Zhang et al., 2025 [4]). However, these methods formulate training data detection primarily as a binary classification problem, failing to provide theoretical guarantees for the set of identified data. Our method is complementary to these heuristic methods, since we can leverage various methods to construct conformal p-values and provide rigorous theoretical guarantees for the identified set.
> - **Unreliable FDR Control**: While KTD attempts to control FDR, it relies on strong distributional assumptions (symmetry of the knockoff statistic) that are often violated in practice, leading to the invalid FDR control shown in Figure 2. Furthermore, KTD requires white-box gradient access, which is often unavailable in copyright disputes involving closed-source models. In contrast, our method achieves strict FDR control with **distribution-free** guarantees, making it applicable in both **white-box and black-box** settings.
>
>
> ### **6. Data Split Setup [Q4]**
>
> Thank you for your suggestions. We have expanded Appendix E.1 in the revised paper by adding a detailed table. Table 3 now details total dataset sizes, member, non-member and calibration counts.
>
> ### **7. Black-box/White-box Terminology [Q5]**
> Thank you for this insightful comment. We respond to the two questions respectively below.
>
> > Does PTDI need access to model outputs?
>
> Yes, our method requires access to the model's outputs for all data points in $\mathcal{D}\_{\text{test}}$  and  $\mathcal{D}\_{\text{cal}}$. The generation of these scores $T(X)$ requires access to the model (either strictly to the output probabilities for black-box scores like Perplexity, or to gradients for white-box scores like Gradient Norm).
>
> > Does PTDI operate identically under a "white-box" regime?
>
> Yes, PTDI would operate mathematically identically regardless of how the score $T(X)$ was generated.
>
> In this work, we use the "black-box" vs. "white-box" terminology primarily to distinguish PTDI from methods like KTD, which explicitly require model gradients. Our method is more versatile as it can be integrated with either black-box or white-box scores. Following your suggestion, we have clarified this terminology in the revised paper.
>
>
> ### **8. Explanation on FDR trends in Figure 2 [Q6]**
> Thank you for this observation. The "leveling off" of the empirical FDR is an expected behavior and is related to the proportion of true nulls (non-members) in the test set.
>
> - Why does the FDR "levels off"? The FDR is the expected proportion of false discoveries. With a higher enough target FDR $\alpha$ (e.g., $\alpha$=1), our method will reject all hypotheses, thereby leading that FDP is exactly the proportion of non-members in the test set. Your intuition is exactly correct: if $\mathcal{D}_{\text{test}}$ is 50% non-members, the absolute maximum possible FDR is 0.5.
> - Why does the FDR of KTD "levels off" sooner? **KTD levels off sooner because its method fails to control the FDR.** As seen in Figure 2, KTD's empirical FDR rapidly violates the target $\alpha$, which means it rapidly escalate towards the maximum possible FDR.
>
>
>
> ### **9. Writing and Formatting Corrections [Q7,Q8,Q9]**
> Thank you for your thorough reading and constructive feedback. We have incorporated all the suggested corrections in the revision.
>
> - Section Title (Line 367): We have renamed the section title to "Effect of the scaling procedure" to correct the grammatical issue.
> - FDR Formatting: We have standardized the FDR notation to use decimals throughout the paper for consistency.
> - Typo (Line 1119): We have corrected the typo.
>
>
>
> [1] "Membership Inference Attacks Cannot Prove that a Model Was Trained On Your Data." （SaTML，2025）
>
> [2] "Detecting pretraining data from large language models." (ICLR, 2024)
>
> [3] "Membership inference attacks against large vision-language models." (ICLR, 2024)
>
> [4] "Fine-tuning can help detect pretraining data from large language models." (ICLR, 2025)

---

> > ### Comment · Reviewer_cobt · 2025-11-27
> > **On the use of Storey's correction with Conformal p-values**
> >
> > **Follow-up Question 1a:** While I acknowledge the authors' statement that: *"*$\ldots$ Storey's estimator cannot be directly applied to the training data identification scenario because its theoretical framework generally relies on the assumption of **independent p-values** derived from their corresponding hypotheses,"* in *Bates et al (2023),* which slightly predates the Jin and Candes (2023) paper the authors have cited, there is a proof (see Theorem 3) that Storey's correction can be applied to conformal p-values (also shown to be PRDS and hence not independent) for FDR control (see their section 2.3 - A positive result: Storey's correction does not break FDR control). **Hence, if I'm not mistaken, this past work already handles this theoretical innovation that the authors are claiming? And if I am mistaken, would the authors kindly help me understand where I am wrong?**
> >
> > At the very least, it would seem that this Bates et al (2023) is another work that must certainly (/obviously) be included in their references.

---

> > > ### Author Response · Authors · 2025-12-03
> > > **Clarification on Theoretical Contribution**
> > >
> > > The theorems in our manuscript aim to provide theoretical guarantees tailored to the training data identification task, rather than proposing a generic statistical framework. We detail our theoretical contributions below:
> > >
> > > - **Theorems for validating the estimator:** We present Theorems 1 and 4 to demonstrate that our proposed estimators strictly control the FDR in their respective scenarios. Our theorems not only validate the subtraction estimator $\pi_{\text{sub}}$ (recovering the same conclusion as Storey’s correction, albeit via a distinct proof strategy) but also enable the derivation of new estimators like $\pi_{\text{mom}}$. Notably, the $\pi_{\text{mom}}$ utilizes external information (known members) to achieve higher power, which is **not covered by previous works**.
> > >
> > > - **Theorem of power improvement:** As noted by the reviewer, prior works do not provide theoretical guarantees regarding power improvements. We address this by providing Theorem 3 to demonstrate that scaling p-values strictly improves power over using the original p-values, which is adaptive to both of our estimators.
> > >
> > > Moreover, our contribution extends beyond the theorems: (1) we formulate training data identification as a multiple-hypothesis testing problem, thereby providing a feasible path to construct provable guarantees where prior work claimed impossibility; (2) Our method is the first to identify training data with **strict FDR control**. We have updated the related work to clarify the position in the revised version.

---

### Author Response · Authors · 2025-11-24
**General Response**

We thank all the reviewers for their time,  thoughtful feedback, and valuable suggestions. We are encouraged that Reviewers GF9Q and jywB find the problem motivation **meaningful** and the method **straightforward**, **clear**, **principled** and **elegant.**. We are pleased that Reviewer NNQR finds the derivation of the scaled p-value **novel** and commends the method’s **strong calibration**. Reviewer cobt and Reviewer GF9Q also recognize the **extensive experiments** covering both LLMs and VLMs with various scoring methods, which demonstrate the method's **realistic applicability**.  We provide point-by-point responses to all reviewers’ comments and concerns below.

The reviews allow us to strengthen our manuscript, and the changes are summarized below:

- Added Theorem 3 proving power improvement in **Appendix C.4**. [cobt]
- Added supplementary details for the comparison experiment with KTD in **Line 356** and **Appendix F.8**. [cobt]
- Added experiments on reference-model-based baselines in **Line 374-375** and **Appendix F.5**. [NNQR]
- Added experiments on fine-tuning settings in **Line 374-375** and **Appendix F.6**. [NNQR]
- Added Hamming distance evaluation in **Appendix F.7**. [GF9Q]
- Expanded discussion on Storey (2002) in Related Work (**Line 801-804, 812-814**). [cobt]
- Added citation of conformal p-values in **Line 165** and **Line 805-806**. [cobt]
- Added detailed data split statistics table in **Line 1172** and **Appendix E.1**. [cobt]
- Clarified “black-box” and "white-box" terminology in **Line 75-76**. [cobt]
- Renamed the section title to "Effect of the scaling procedure" in **Line 367**
- Fixed typos in **Line 131, 1178** and standardized FDR formatting in **Line 67-70**. [cobt, GF9Q]

¹ For clarity, we highlight the revised part of the manuscript in **blue** color.

---

### Author Response · Authors · 2025-12-03
**Summary of Rebuttal**

Dear Area Chair,

Thank you for handling our submission following the OpenReview incident. We sincerely appreciate your efforts in overseeing our submission and managing the significant additional load during this period.

For clarification, we highlight the contributions and novelty of our work as follows: (1) formulating training data identification as a multiple-hypothesis testing problem, thereby providing a feasible path to construct provable guarantees where prior work claimed impossibility; (2) introducing the first method to identify training data with a valid theoretical guarantee, without requiring reference models and gradients; and (3) establishing rigorous theorems that guarantee strict FDR control for both proposed estimators across distinct scenarios, and providing theoretical analysis for the power improvement.

Below, we outline how our rebuttal addresses the reviewer’s primary concerns:

1. **Reviewer cobt** raised concerns regarding the **theoretical contributions**, as well as the **comparison with KTD**. In response:
    - **Theoretical Contributions:** We clarified our contribution is providing theoretical guarantees tailored to the training data identification task. In particular, we (a) established **Theorems 1 & 4** to demonstrate that our proposed estimators strictly control the FDR in their respective scenarios; and (b) provided **Theorem 3** to confirm that our scaling p-values strategy achieves higher power than the original p-values.

     - **Comparison with KTD:** We clarified that power comparisons are only valid when FDR is controlled. As shown in the expanded results in Appendix F.8, unlike KTD, which fails to strictly control FDR, PTDI is the only method that both strictly controls FDR across all settings and achieves superior power.`


2. **Reviewer NNQR** found the derivation **novel** and the method **strongly calibrated** but requested **validation across broader settings** and justification for the power improvement. In response:
    - **Generalization to New Settings:** We added experiments using reference-based scores (Ref-Small, LiRA) and fine-tuning settings (**Appendix F.5 & F.6**). Our method consistently improves power over the vanilla baseline in these settings, demonstrating versatility beyond pre-training detection.
    -  **Reason for Power Improvement:** We provided the theoretical justification (**Theorem 3**) clarifying that our method improves power by optimizing the decision threshold while maintaining strict FDR control, rather than changing the scoring ranking.


3. **Reviewer GF9Q** found the problem **meaningful** but mainly questioned the realism of the **data assumptions**. Moreover, this reviewer expressed willingness to increase the score prior to the rebuttal. In response:
    - **Data Distribution Assumptions:** We clarified that our method relies on the *marginal* distribution of scores rather than assuming identical detection behavior across samples. We also clarified that our method estimates the proportion for the *given* test set, **without assumption for uniform sampling**.



4. **Reviewer jywB** assessed the method as **clear, principled, and elegant**, with a specific query regarding the **circularity** of the adjusted moment estimator. In response:
    - **Clarification on the Adjusted Moment Estimator:** We clarified that the adjusted moment estimator ($\pi_{\text{mom}}$) utilizes **disjoint** public corpora (e.g., The Pile) as confirmed training samples. This is a standard auditing practice and ensures there is no circular logic in the estimation process.


We believe these revisions have significantly strengthened the manuscript, establishing it as the first effective method to provide provable statistical guarantees for training data identification in large-scale models. We hope this summary assists your final assessment, and we sincerely appreciate the constructive engagement from both the reviewers and the AC.

Best,

The Authors

---

### Meta-Review · Area_Chair_wN91 · 2026-01-05

**Summary:**

The paper proposes a new method for training data inference through the lens of multiple hypothesis testing. Reviewers were optimistic about the problem formulation and the elegance of the method. However, there was also some concern about the novelty of the theoretical derivation (esp. by reviewer cobt w.r.t. prior work by Bates et al) and of the intuition for where the gains are coming from (by reviewer GF9Q) that seem strongly tied to the knowledge of side information re: data usage proportion. Overall, I'm optimistic about the message of the paper, but I don't think there is enough enthusiasm or clarity on the novelty concerns to accept the paper.

I think the work has merit and could benefit from a revision, where I suggest that the authors (1) cite the Bates et al paper and clearly demarcate differences in their main theoretical results (is it primarily about compatibility with an estimated member proportion?) and (2) if the primary novelty is in formulating membership inference as a multiple hypothesis testing problem (as suggested by the "Summary of Rebuttal"), rather than the methods / theorems themselves, then reposition the presentation around this. I like that this formulation bypasses impossibility results from Zhang et al, and clearly exposing these merits while clarifying novelty may improve the paper.

**Reviewer Concerns:**

Many of the concerns about the problem formulation and assumptions seem adequately addressed to me. The main outstanding concern is by reviewer cobt, who asked about the relationship between the main theorems and prior work of Bates et al; it seems from my reading of the rebuttal that there is a great deal of conceptual overlap (perhaps with a different proof technique), and the main innovation is in deriving other consequences, e.g., in presence of an estimated data usage rate.

**Reviewer Scores:**

I suspect that cobt may be left unconvinced by the technical merits based on the existing discussion. Because cobt gave the lowest score and highest confidence, and the only "accept" rating was relatively sparse, I can't recommend a higher rating with the current reviews.

---

### Decision · Program_Chairs · 2026-01-26

Reject